# Learning to Generalize: An Information Perspective on Neural Processes

**Hui Li**[1,2,3,†], **Huafeng Liu**[2,3,†], **Shuyang Lin**[1,2,3], **Jingyue Shi**[1,2,3],

**Yiran Fu**[1,2,3], **Liping Jing**[1,2,3,*]

[1]State Key Laboratory of Advanced Rail Autonomous Operation, Beijing, China
[2]School of Computer Science and Technology, Beijing Jiaotong University, Beijing, China
[3]Beijing Key Laboratory of Traffic Data Mining and Embodied Intelligence, Beijing, China
`{huili97, hfliu1, sylin1, jingyueshi, yiranfu, lpjing}@bjtu.edu.cn`

## Abstract

Neural Processes (NPs) combine the adaptability of neural networks with the efficiency of meta-learning, offering a powerful framework for modeling stochastic processes. However, existing methods focus on empirical performance while lacking a rigorous theoretical understanding of generalization. To address this, we propose an information-theoretic framework to analyze the generalization bounds of NPs, introducing dynamical stability regularization to minimize sharpness and improve optimization dynamics. Additionally, we show how noise-injected parameter updates complement this regularization. The proposed approach, applicable to a wide range of NP models, is validated through experiments on classic benchmarks, including 1D regression, image completion, Bayesian optimization, and contextual bandits. The results demonstrate tighter generalization bounds and superior predictive performance, establishing a principled foundation for advancing generalizable NP models.

## 1 Introduction

Gaussian processes (GPs) are widely recognized as a robust framework for modeling distributions over functions [37]. Their appeal lies in the consistent probabilistic reasoning enabled by Bayesian inference, which facilitates data-efficient modeling. Despite their strengths, GPs are unsuitable for certain problems. For instance, a function exhibiting a single, unknown discontinuity is a classic example of a distribution that GPs fail to represent [34].

To address such limitations, researchers have turned to neural network-based generative models. Notable advancements in this area include meta-learning techniques like Neural Processes (NPs) [14, 15] and models based on variational autoencoders (VAEs) [33, 11]. These approaches leverage extensive small dataset training to transfer knowledge effectively across tasks during prediction. Neural networks (NNs) offer additional advantages by offloading computational intensity to the training phase, simplifying predictions, and freeing the model from Gaussianity constraints.

Building upon the foundational work of Conditional Neural Processes (CNPs) and Neural Processes (NPs) [14, 15], numerous studies have enhanced NPs by integrating advanced mechanisms. Attentive Neural Processes (ANPs) [27] incorporated attention mechanisms to better model long-range dependencies, while Transformer Neural Processes (TNPs) [36] leveraged self-attention for improved scalability. Convolutional Neural Processes (ConvCNPs) [10] adapted convolutional architectures to excel in spatial data tasks, and Neural Diffusion Processes (NDPs) [7] introduced diffusion mechanisms to enhance uncertainty estimation. NPCL [22] extended NPs to continual learning, enabling

---

*Corresponding authors: Liping Jing; †These authors contributed equally to this work. Codes: `https://github.com/Allen0497/Gen-NPs`

sequential task adaptation, and multimodal approaches like [25] generalized NPs for uncertainty estimation across multiple modalities.

While these advancements significantly enhance the experimental performance of NPs, they primarily focus on task-specific improvements through architectural innovations and meta-learning strategies [20, 16]. However, the generalization capability of NPs—a critical hallmark of their design—has received limited theoretical exploration. Existing works prioritize performance optimization, often neglecting the theoretical interpretability of NPs. For instance, many methods emphasize improving inductive biases or increasing model complexity without addressing the theoretical underpinnings of NPs' generalization ability [17, 10]. This gap underscores the need for a formal theoretical foundation to better understand and generalize NPs across broader domains and tasks, paving the way for more principled and interpretable neural process models [44].

Information theory provides a powerful framework for analyzing and quantifying uncertainty, offering rigorous tools to evaluate the generalization of models [18, 35]. This paper introduces an information-theoretic framework to analyze and improve the generalization bounds of Neural Processes (NPs). Building on this analysis, we propose a novel approach that incorporates dynamical stability regularization, which explicitly minimizes sharpness through the trace and Frobenius norm of the Hessian matrix. Additionally, we demonstrate how noise-injected parameter updates complement this regularization by smoothing the optimization trajectory and improving gradient coherence. These insights are broadly applicable across NP variants and validated through extensive experiments on benchmarks such as 1D regression, image completion, Bayesian optimization, and contextual bandits.

The main contributions of this paper are as follows:

- We conduct an information-theoretic analysis of NP generalization bounds, providing a rigorous perspective on their generalization capabilities.

- We propose a dynamical stability regularization framework, complemented by noise-injected parameter updates, to enhance generalization by minimizing sharpness and improving optimization dynamics.

- Extensive experiments on classic NP tasks validate our approach, demonstrating tighter generalization bounds and superior performance compared to existing methods.

## 2 Related Work

In this section, two relevant areas are briefly reviewed: neural processes and information-theoretic learning.

**Neural Processes** Neural Processes (NPs) are probabilistic models designed to meta-learn distributions over functions, offering a data-driven alternative to Gaussian Processes (GPs) [14, 15]. Unlike GPs, which rely on manually specified priors, NPs employ an encoder-decoder architecture to learn a family of functions, efficiently capturing uncertainty using neural networks. Conditional Neural Processes (CNPs) [14], the earliest variant, introduced a deterministic encoder-decoder framework but assumed independence among predictive outputs. Latent Neural Processes (NPs) [15] addressed this by introducing a global latent variable for uncertainty modeling. Subsequent works have enhanced expressiveness, scalability, and uncertainty modeling. Extensions of NPs have addressed specific limitations and broadened their applicability. Attentive Neural Processes (ANPs) [27] incorporated attention mechanisms to better model long-range dependencies and heteroscedastic uncertainty. Convolutional Neural Processes (ConvCNPs) [10] introduced translation-equivariant architectures, excelling in spatial and structured data tasks. Transformer Neural Processes (TNPs) [36] leveraged self-attention for scalability and sequence modeling, while Neural Diffusion Processes (NDPs) [7] used diffusion mechanisms to improve robustness and uncertainty estimation. Neural Processes for Uncertainty-Aware Continual Learning (NPCL) [22] extended NPs to sequential task adaptation without catastrophic forgetting. Additionally, multimodal extensions [25] generalized NPs to handle multi-sensor fusion and uncertainty estimation. These advancements highlight the versatility of NPs and their adaptability across diverse applications.

**Information-Theoretic Learning** Information-theoretic learning (ITL) leverages principles from information theory to analyze and optimize learning algorithms. Unlike traditional approaches based on measures like VC-dimension [39] or uniform stability [4], ITL characterizes learning using mutual information and related quantities [50, 38]. By framing generalization error as the mutual information between inputs and outputs, ITL captures dependencies among data distribution, hypothesis space, and learning algorithms, addressing challenges in uncertainty quantification, robustness, and data

efficiency. This framework provides valuable insights into the generalization capabilities of modern machine learning models. Recent advances have expanded ITL's application to generalization analysis and enhancement. Harutyunyan et al. [18] proposed improved generalization bounds by focusing on the mutual information between predictions and the training set, offering practical estimates of the generalization gap in deep learning. Neu et al. [35] extended ITL to stochastic gradient descent (SGD), deriving tighter bounds based on local gradient statistics and sensitivity along the optimization path. These advances demonstrate ITL's utility in understanding complex learning dynamics and improving theoretical guarantees for generalization.

## 3 Preliminaries

This section introduces the foundational concepts and mathematical tools necessary for understanding the methods and analyses presented in this paper, including information theory, neural processes, and dynamical stability.

**Information Theory**  Let $P_X$ denote the marginal distribution of the random variable $X$. For the Markov chain $X \to Y$, the conditional distribution (or Markov transition kernel) is denoted by $P_{Y|X}$, and the notation $X \perp Y$ indicates independence between $X$ and $Y$ [6]. The cumulant generating function (CGF) of a random variable $X$ is defined as $\psi_X(\lambda) \triangleq \log \mathbb{E}[e^{\lambda(X-\mathbb{E}[X])}]$, where $\lambda$ is a real number. A random variable $X$ is $\sigma$-subgaussian if its CGF satisfies $\psi_X(\lambda) \leq \frac{\lambda^2 \sigma^2}{2}$ for all $\lambda \in \mathbb{R}$ [43].

The mutual information between $X$ and $Y$ is defined as $I(X; Y) \triangleq KL(P_{X,Y} \| P_X P_Y)$, where $P_{X,Y}$ is the joint distribution of $X$ and $Y$, and $P_X P_Y$ is the product of their marginals [6]. The disintegrated mutual information between $X$ and $Y$ given $U$ is $I^U(X; Y) \triangleq KL(P_{X,Y|U} \| P_{X|U} P_{Y|U})$, where $P_{X,Y|U}$ is the conditional joint distribution given $U$. The conditional mutual information is then defined as $I(X; Y|U) \triangleq \mathbb{E}_U[I^U(X; Y)]$, the expectation of the disintegrated mutual information over the distribution of $U$ [1].

**Neural Processes**  Neural Processes (NPs) model the conditional predictive distribution of target values $\mathbf{y}^T$ at target points $\mathbf{X}^T$ based on a context set $\mathcal{D}^C$, expressed as $P(\mathbf{y}^T|\mathbf{X}^T, \mathcal{D}^C)$ [15]. For deterministic NPs (CNPs), the conditional distribution is simplified as $P(\mathbf{y}^T|\mathbf{X}^T, \mathcal{D}^C) = P(\mathbf{y}^T|\mathbf{X}^T, \mathbf{r}^C)$, where $\mathbf{r}^C$ is an aggregated feature of $\mathcal{D}^C$ [14]. Probabilistic NPs introduce a latent variable $\mathbf{z}$ to capture uncertainty, modeling $P(\mathbf{y}^T|\mathbf{X}^T, \mathcal{D}^C) = \int P_\theta(\mathbf{y}^T|\mathbf{X}^T, \mathbf{z}) P_\theta(\mathbf{z}|\mathcal{D}^C) \, d\mathbf{z}$. Training maximizes the evidence lower bound (ELBO): $\mathbb{E}_{\mathbf{z} \sim P_\theta(\mathbf{z}|\mathcal{D}^C)}[\log P(\mathbf{y}|\mathbf{X})] - KL[P_\theta(\mathbf{z}|\mathbf{X}, \mathcal{D}^C)\|P_\theta(\mathbf{z}|\mathcal{D}^C)]$ [15]. During meta-training, NPs learn from tasks sampled from an environment $\tau$, a probability measure over task distributions $\mu$, where each task involves dividing data into context and target sets [20]. A meta-dataset consists of $m$ datasets $\mathcal{D}_{1:m} = (\mathcal{D}_1, \ldots, \mathcal{D}_m)$, each sampled independently from $\mu_{n,\tau}$, the mixture distribution induced by $\tau$ [9]. The meta learner $\mathcal{A}$ outputs a meta-parameter $\theta = \mathcal{A}(\mathcal{D}_{1:m}) \sim P_{\theta|\mathcal{D}_{1:m}}$, representing shared knowledge across tasks [3]. The meta risk is defined as $R_\tau(\theta) \triangleq \mathbb{E}_{\mathcal{D} \sim \mu_{n,\tau}, \mu \sim \tau}[\mathbb{E}_{\mathcal{D}^C}[R_\mu(\theta)]]$, where $R_\mu(\theta) \triangleq -\mathbb{E}_{(x,y) \sim \mu} \log P(y|x, \mathcal{D}^C)$ [3]. The empirical meta risk is given by $R_{\mathcal{D}_{1:m}}(\theta) \triangleq \frac{1}{m} \sum_{j=1}^m R_{\mathcal{D}_j}(\theta)$, with $R_{\mathcal{D}_j}(\theta) \triangleq -\frac{1}{n} \sum_{i=1}^n \log P(y_i|x_i, \mathcal{D}_j^C)$. The meta generalization error is then defined as $\text{gen}_{meta}^{NPs}(\tau, \mathcal{A}) \triangleq \mathbb{E}_{\theta, \mathcal{D}_{1:m}}[R_\tau(\theta) - R_{\mathcal{D}_{1:m}}(\theta)]$, which quantifies the difference between true and empirical meta risks [3]. By learning a meta-parameter $\theta$ shared across tasks, NPs efficiently adapt to new tasks, leveraging shared information in the environment.

**Dynamical Stability**  Dynamical stability refers to the robustness of the stochastic gradient descent (SGD) optimization process to small perturbations in the parameter space and has been shown to play a critical role in both optimization and generalization [49, 42]. Consider the parameter update rule $\theta_{s+1} = \theta_s - \eta \nabla R_s(\theta_s)$, where $\eta$ is the learning rate, and $\nabla R_s(\theta_s)$ is the stochastic gradient. The sensitivity of the optimization dynamics can be quantified by the spectral norm $\|J_s\|_2$ of the Jacobian matrix $J_s = \frac{\partial \theta_{s+1}}{\partial \theta_s}$, which governs how small perturbations $\delta_s$ at step $s$ propagate through subsequent iterations [48, 40, 47]. When $\|J_s\|_2 < 1$, perturbations decay over time, leading to stable training. Importantly, this stability imposes an implicit regularization effect, as SGD inherently biases the optimization process towards flatter minima with lower curvature, measured by the Hessian $H = \nabla^2 R(\theta)$ [21, 26]. Flatter minima are associated with better generalization due to their robustness to noise and parameter variations [19, 8]. Moreover, the Lyapunov exponents, which quantify the exponential rates of divergence or convergence of nearby trajectories in parameter space, provide a formal link between stability and generalization. Negative Lyapunov exponents imply stable dynamics and a tendency towards generalizable solutions [42, 49]. This perspective highlights the central role of dynamical stability in understanding the implicit biases of SGD and its impact on both the optimization landscape and model generalization [47, 41].

# 4 Methodology

This section investigates the generalization properties of Neural Processes (NPs) by integrating information-theoretic principles with optimization dynamics. First, we quantify the generalization error of NPs using mutual information (MI), capturing uncertainties from data and task distributions. Next, we introduce a risk-aware dynamical stability regularization term $R_{\text{dyn}}$, addressing sharpness and curvature to improve generalization. Finally, we propose an optimization-aware noise injection strategy to enhance stability and guide parameter updates toward flatter minima.

## 4.1 Quantifying Generalization with Information Theory

Understanding the generalization capabilities of Neural Processes (NPs) requires quantifying the uncertainties inherent in the learning process. By leveraging mutual information (MI), the generalization error can be decomposed into components reflecting different sources of uncertainty, offering insights into the behavior of NPs across diverse tasks. While MI provides a rigorous framework to analyze generalization, standard approaches often neglect the impact of model complexity, which is particularly critical in the hierarchical nature of meta-learning. This section lays the foundation for incorporating complexity constraints in future analysis.

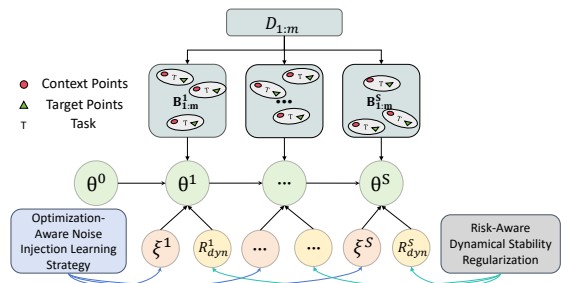

Figure 1: Generalization error of Neural Processes (NPs) varies with the Hessian trace

To analyze generalization bounds, we consider the relationship between true risk and empirical risk under a given task distribution. Let $\mu$ represent an unknown distribution on the instance space $X \times Y$. A meta-dataset consists of $m$ datasets, denoted as $\mathcal{D}_{1:m} = (\mathcal{D}_1, \ldots, \mathcal{D}_m)$, where each dataset $\mathcal{D}_j$ consists of $n$ independent samples drawn from $\mu$. The full dataset is thus $\mathcal{D} = \mathcal{D}_{1:m} = \{\mathcal{D}_j\}_{j=1}^m$. For a hypothesis $\theta$, the true risk is defined as $R_\mu(\theta) \triangleq \mathbb{E}_{X \times Y \sim \mu}[\ell(\theta, X \times Y)]$, and the empirical risk as $R_\mathcal{D}(\theta) \triangleq \frac{1}{mn} \sum_{j=1}^m \sum_{i=1}^n \ell(\theta, x_{i,j}, y_{i,j})$, where $\ell : \Theta \times X \times Y \to \mathbb{R}$ is the loss function. A learning algorithm $\mathcal{A}$ maps the dataset $\mathcal{D}$ to a randomized hypothesis $\theta = \mathcal{A}(\mathcal{D}) \sim P_{\theta|\mathcal{D}}$, and its generalization error is defined as $\text{gen}(\mu, \mathcal{A}) \triangleq \mathbb{E}_{\theta, \mathcal{D}}[R_\mu(\theta) - R_\mathcal{D}(\theta)]$.

**Theorem 4.1.** *If the loss $\ell(\theta, X \times Y)$ is $\sigma$-subgaussian for each $\theta \in \Theta$ with respect to $X \times Y \sim \mu$, the generalization error of a learning algorithm $\mathcal{A}$ satisfies the bound $|gen(\mu, \mathcal{A})| \leq \sqrt{\frac{2\sigma^2}{mn} I(\theta; \mathcal{D})}$, where $I(\theta; \mathcal{D})$ is the mutual information between the dataset $\mathcal{D}$ and the hypothesis $\theta$. For meta-learning tasks, where $\mu \sim \tau$, the meta-generalization error satisfies:*

$$\left| gen_{meta}^{NPs}(\tau, \mathcal{A}) \right| \leq \sqrt{\frac{2\sigma^2}{mn} I(\theta; \mathcal{D}_{1:m})}. \tag{1}$$

*The mutual information $I(\theta; \mathcal{D}_{1:m})$ can be computed based on the joint distribution $P(\theta, \mathcal{D}_{1:m})$ and the marginal $P(\theta)$; detailed derivations are provided in the appendix. This bound indicates that minimizing the dependence of $\theta$ on the data (via $I(\theta; \mathcal{D}_{1:m})$) improves the generalization performance.*

Although the MI framework elegantly quantifies generalization, it does not explicitly account for the hypothesis space complexity or the solution sharpness. For NPs, these factors are especially significant as a result of their hierarchical structure. The hypothesis space in NPs, determined by the meta-parameters $\theta$, encodes both task-level priors and task-specific adaptations. This space is inherently high-dimensional, making it prone to overfitting, particularly when the meta-dataset $\mathcal{D}_{1:m}$ is small or lacks diversity.

Figure 2: The training process of Gen-NPs

The sharpness of the learned solution is related to the curvature of the loss surface, quantified by the Hessian $H = \nabla^2 \mathcal{R}(\theta)$. A high Hessian

trace implies sensitivity to parameter perturbations, adversely affecting generalization, especially across diverse task distributions. As illustrated in Figure 1, the generalization error of NPs deteriorates with increasing Hessian trace.

Based on the above analysis, we focus on improving NPs' generalization from both risk-aware dynamical stability regularization and optimization-aware noise injection learning strategy, the whole learning framework is given in Figure 2.

## 4.2 Risk-Aware Dynamical Stability Regularization

Inspired by previous works that demonstrate the critical role of sharpness and curvature in optimization and generalization [26, 23, 45], we introduce a dynamical stability regularization (DSR) term $R_{\text{dyn}}$ that incorporates model complexity constraints into the generalization bound. Specifically, sharpness of the solution has been linked to poor generalization, motivating the need to explicitly regularize the curvature of the loss landscape [19, 21, 51]. Moreover, recent studies have explored how Lyapunov stability and noise in optimization dynamics can implicitly bias solutions towards flat minima, further supporting the necessity of stability-based regularization [40, 49].

**Definition 4.2.** The dynamical stability regularization term $R_{\text{dyn}}$ quantifies the complexity of the hypothesis space by leveraging properties of the Hessian matrix $H$, which represents the second-order derivatives of the loss function with respect to the parameters $\theta$:

$$R_{\text{dyn}} = \lambda_1 \cdot \mathbb{E}[Tr(H)] + \lambda_2 \cdot \mathbb{E}[\|H\|_F], \tag{2}$$

where $Tr(H)$ (the trace of the Hessian) measures the solution sharpness by summing the eigenvalues of $H$, and $\|H\|_F$ (the Frobenius norm) captures the overall curvature of the loss landscape. The hyperparameters $\lambda_1$ and $\lambda_2$ control the relative importance of these two components, with empirical values typically set as $\lambda_1 \in [0.01, 0.1]$ and $\lambda_2 \in [0.001, 0.01]$ to maintain an appropriate balance between the two terms. By penalizing sharpness and curvature, $R\text{dyn}$ aligns the theoretical bounds with practical observations in optimization.

**Theorem 4.3.** *By incorporating $R_{dyn}$ into the information-theoretic framework, the refined meta-generalization error bound for Neural Processes (NPs) is given by:*

$$\left| gen_{meta}^{NPs}(\tau, \mathcal{A}_{DSR}) \right| \leq \sqrt{\frac{2\sigma^2}{nm} \frac{I(\theta; \mathcal{D}_{1:m})}{1 + \alpha \cdot R_{dyn}}}, \tag{3}$$

*where $I(\theta; \mathcal{D}_{1:m})$ is the mutual information between the meta-parameter $\theta$ and the dataset $\mathcal{D}_{1:m}$, $R_{dyn}$ is the dynamical stability regularization term, and $\alpha > 0$ is a scaling factor that controls the influence of $R_{dyn}$. The proof of this theorem is provided in the Appendix.*

Including $R_{\text{dyn}}$ in the generalization bound has notable effects. Penalizing high $Tr(H)$ encourages the model to select flatter minima, which are associated with better generalization. Additionally, constraining the Frobenius norm $\|H\|_F$ prevents overfitting to high-curvature regions. This regularization complements the implicit bias of optimization algorithms, such as gradient descent, which naturally steer solutions toward regions of lower sharpness.

The role of $R_{\text{dyn}}$ depends on the training regime. When the number of tasks $m$ is large ($m \rightarrow \infty$) with fixed samples per task $n$, the mutual information term $I(\theta; \mathcal{D}_{1:m})$ dominates the bound, and $R_{\text{dyn}}$ constrains the sharpness across tasks. Conversely, when $n$ grows ($n \rightarrow \infty$) while $m$ remains fixed, the generalization error decreases naturally as task-level uncertainty reduces, making $R_{\text{dyn}}$ less critical.

From a practical perspective, incorporating $R_{\text{dyn}}$ into the generalization bound provides actionable insights for optimizing Neural Processes. Maximizing $R_{\text{dyn}}$ during training improves generalization, and tuning the hyperparameters $\lambda_1$ and $\lambda_2$ balances sharpness reduction and complexity control. Ultimately, $R_{\text{dyn}}$ bridges the gap between theoretical generalization bounds and practical challenges, leading to tighter bounds and a deeper understanding of the factors influencing generalization in NPs.

## 4.3 Optimization-Aware Noise Injection Learning Strategy

Motivated by the well-established relationship between noise injection and generalization [19, 41, 46], we propose an optimization-aware noise injection strategy tailored for meta-learning tasks. Noise has been shown to play a critical role in escaping sharp minima [26], smoothing the loss landscape [21], and improving generalization through implicit regularization [47]. Furthermore, recent studies highlight the importance of integrating noise injection into gradient-based optimization to ensure stability and guide parameter trajectories towards flatter regions of the loss landscape [24, 49].

Building upon these insights, we explicitly incorporate noise into the parameter update rule to enhance both the stability and generalization of meta-parameter optimization. The overall training process of NPs, incorporating noise injection and dynamical stability regularization, is illustrated in Figure 2. The pseudocode for the entire algorithm can be found in the Appendix.

To explicitly incorporate the effect of noise into the learning dynamics, we modify the standard gradient-based parameter update rule by introducing an isotropic Gaussian noise term $\xi^s$ at each iteration $s$. The parameter update rule for meta-parameters $\theta$ at iteration $s$ is formally given by:

$$\theta^s = \theta^{s-1} - \eta_s \nabla[\tilde{R}_{\mathcal{D}_i^T}(\theta^{s-1}) + R_{\text{dyn}}] + \xi^s, \tag{4}$$

where $\tilde{R}_{\mathcal{D}_i^T}(\theta^{s-1}) + R_{\text{dyn}}$ is the empirical risk computed on the target set $\mathcal{D}_i^T$ of task $i$, $\eta_s$ is the learning rate at iteration $s$, and $\xi^s \sim \mathcal{N}(0, \sigma_s^2 I_k)$ is an isotropic Gaussian noise term with variance $\sigma_s^2$ and dimensionality $k$. This update rule consists of two key components: the gradient descent step, which minimizes the empirical risk, and the noise injection step, which perturbs the parameter updates. The variance $\sigma_s^2$ can be adjusted over iterations to control the strength of the perturbation.

The introduction of noise into the parameter updates serves two primary purposes. First, it provides implicit regularization by encouraging the model to explore flatter regions of the loss landscape and favor minima with lower sharpness. This aligns with the objective of maximizing $R_{\text{dyn}}$, as described in Section 4. Second, noise facilitates escaping sharp minima, which are common in high-dimensional parameter spaces and often lead to poor generalization. By perturbing the trajectory of the updates, noise helps the optimization process avoid these undesirable regions.

To achieve the desired regularization effect without destabilizing the optimization, the variance $\sigma_s^2$ of the noise is carefully scaled with respect to the learning rate $\eta_s$. A common choice for this scaling is $\sigma_s^2 = \eta_s/\gamma$, where $\gamma > 0$ is a scaling factor that controls the relative strength of the noise. This formulation ensures that the noise magnitude decreases as the learning rate decreases, allowing the optimization to stabilize in later iterations.

The noise-injected parameter update rule in Eq. 4 directly influences the dynamical stability regularization term $R_{\text{dyn}}$ by affecting the Hessian properties of the loss landscape. Specifically, the perturbation introduced by $\xi^s$ reduces the likelihood of converging to regions with high $Tr(H)$, thus implicitly maximizing the sharpness of the solution. Furthermore, noise smooths the optimization trajectory, reducing the overall curvature of the solution as measured by $\|H\|_F$. Together, these effects demonstrate the critical role of noise injection in improving both the stability and generalization performance of meta-parameter optimization. While introducing noise and regularization incurs additional computational overhead, the performance gains justify these costs (detailed analysis in Appendix).

## 5 Deeper Analysis

In this section, we establish a theoretical foundation for understanding how noise and dynamical stability regularization affect the generalization performance of NPs. By analyzing the interplay between noise, sharpness, and mutual information, we demonstrate their influence on generalization bounds. Additionally, we show how noise injection mitigates gradient incoherence, a key challenge in meta-learning, to improve stability and generalization.

### 5.1 Theoretical Analysis

We analyze how noise-injected parameter updates affect the dynamical stability of the optimization process and its impact on generalization performance. Specifically, we focus on their effects on sharpness, curvature (captured by the dynamical stability regularization term $R_{\text{dyn}}$), and mutual information $I(\theta; \mathcal{D}_{1:m})$. This analysis highlights the role of noise in reducing both $R_{\text{dyn}}$ and $I$, thereby improving the generalization of NPs.

To begin, we linearize the parameter updates around the optimal solution $\theta^*$, which minimizes the true risk $R_\mu(\theta)$. Let $\tilde{\theta}^s = \theta^s - \theta^*$ represent the deviation from $\theta^*$ at iteration $s$. The linearized noise-injected parameter update rule is:

$$\tilde{\theta}^{s+1} = (I - \eta_s H)\tilde{\theta}^s + \xi^s, \tag{5}$$

where $H = \nabla^2[\tilde{R}_{\mathcal{D}_i^T} + R_{\text{dyn}}](\theta^*)$ is the Hessian of the empirical risk, and $\xi^s \sim \mathcal{N}(0, \sigma_s^2 I_k)$ is isotropic Gaussian noise. This form separates the contributions of optimization dynamics $(I - \eta_s H)$ and noise $\xi^s$ to the evolution of $\tilde{\theta}^s$.

Noise affects sharpness, measured by the trace of the Hessian $Tr(H)$, by perturbing the optimization trajectory and encouraging exploration of flatter regions. This perturbation has a dual effect on the loss landscape: it locally reduces the curvature around specific minima while simultaneously increasing the system's overall stability by promoting robust, flat solutions. Consequently, the effective Hessian trace in the presence of noise satisfies $Tr(H_{\text{noise}}) \geq Tr(H) \cdot (1 + \eta_s \sigma_s^2)$, where a higher variance of noise $\sigma_s^2$ results in a greater enhancement of dynamic stability. Similarly, the Frobenius norm $\|H\|_F$, which provides a comprehensive measure of the curvature of the loss landscape, satisfies $\|H_{\text{noise}}\|_F \geq \|H\|_F \cdot \sqrt{1 + \eta_s \sigma_s^2}$, indicating that noise enhances the system's ability to maintain stability in the face of parameter perturbations. These results suggest that noise injection encourages the optimization process to explore flatter regions of the loss landscape, develop higher dynamic stability, and improve generalization performance by enhancing robustness to parameter perturbations.

By enhancing both stability metrics, noise effectively amplifies the dynamical stability regularization term $R_{\text{dyn}}$, which appears in the denominator of our generalization bound. Specifically, substituting the enhanced trace and Frobenius norm into the definition of $R_{\text{dyn}}$ yields:

$$R_{\text{dyn, noise}} \geq \lambda_1 \cdot \mathbb{E}[Tr(H)] \cdot (1 + \eta_s \sigma_s^2) + \lambda_2 \cdot \mathbb{E}[\|H\|_F] \cdot \sqrt{1 + \eta_s \sigma_s^2} \tag{6}$$

This increase in $R$dyn directly tightens the generalization bound by increasing the denominator term $1 + \alpha \cdot R_{\text{dyn}}$, resulting in improved generalization capabilities for Neural Processes.

As the noise variance $\sigma_s^2$ decreases over iterations ($\sigma_s^2 = \eta_s/\gamma$), the optimization stabilizes near a flat minimum, with the stability benefits retained as enhanced $R_{\text{dyn}}$. Noise also reduces the mutual information $I(\theta; \mathcal{D}_{1:m})$, which quantifies the dependency between model parameters $\theta$ and training data $\mathcal{D}_{1:m}$. High mutual information indicates potential overfitting, as parameters become overly dependent on specific data. The reduction is given by:

$$I_{\text{noise}}(\theta; \mathcal{D}_{1:m}) \leq \max\left(0, I(\theta; \mathcal{D}_{1:m}) - \eta_s \sigma_s^2 \cdot \mathbb{E}[\|\nabla \tilde{R}_{\mathcal{D}_i^T}(\theta)\|^2]\right). \tag{7}$$

This ensures $I_{\text{noise}}(\theta; \mathcal{D}_{1:m})$ remains non-negative and significantly lower than the original mutual information. Excessive noise or high learning rates, however, may destabilize optimization, limiting these benefits. By jointly reducing $I(\theta; \mathcal{D}_{1:m})$ and amplifying $R_{\text{dyn}}$ through increased $Tr(H)$ and $\|H\|_F$, noise-injected updates tighten the generalization bound:

$$\left|\text{gen}_{\text{meta}}^{\text{Gen-NPs}}(\tau, \mathcal{A})\right| \leq \sqrt{\frac{2\sigma^2}{nm} \frac{I_{\text{noise}}(\theta; \mathcal{D}_{1:m})}{1 + \alpha \cdot R_{\text{dyn, noise}}}} \leq \sqrt{\frac{2\sigma^2}{nm} \frac{I(\theta; \mathcal{D}_{1:m})}{1 + \alpha \cdot R_{\text{dyn}}}}, \tag{8}$$

where the second inequality follows from $I_{\text{noise}}(\theta; \mathcal{D}_{1:m}) \leq I(\theta; \mathcal{D}_{1:m})$ and $R_{\text{dyn, noise}} \geq R_{\text{dyn}}$. This interplay between noise, mutual information, and dynamical stability is crucial for achieving tighter generalization bounds and improving meta-learning performance.

### 5.2 Gradient Incoherence

Gradient incoherence, caused by inconsistencies between gradients computed on the context set ($\mathcal{D}_i^C$) and the target set ($\mathcal{D}_i^T$), affects the stability and generalization of Neural Processes (NPs). This issue can lead to suboptimal updates and hinder learning. Noise-injected parameter updates mitigate this problem by smoothing the loss landscape and reducing gradient mismatches.

**Definition 5.1.** The gradient incoherence for a task $i$ at iteration $s$ is defined as the $\ell_2$-norm of the difference between the gradients on the full dataset $\mathcal{D}_i$ and the target set $\mathcal{D}_i^T$:

$$\epsilon_{i,s}^{\theta} = \left\| \nabla[\tilde{R}_{\mathcal{D}_i}(\theta^{s-1}) + R_{dyn}] - \nabla[\tilde{R}_{\mathcal{D}_i^T}(\theta^{s-1}) + R_{dyn}] \right\|_2^2, \tag{9}$$

where $\nabla\tilde{R}_{\mathcal{D}_i}(\theta^{s-1}) + R_{dyn}$ and $\nabla\tilde{R}_{\mathcal{D}_i^T}(\theta^{s-1}) + R_{dyn}$ are gradients of the empirical risk over $\mathcal{D}_i$ and $\mathcal{D}_i^T$, respectively. The overall gradient incoherence across tasks and iterations is given by GI $= \frac{1}{mS}\sum_{i=1}^m \sum_{s=1}^S \epsilon_{i,s}^{\theta}$, where $m$ is the number of tasks and $S$ the number of iterations. Minimizing GI ensures more consistent updates, improving generalization.

Noise reduces gradient incoherence by perturbing the optimization trajectory. The noise $\xi^s$ introduced during updates disrupts gradient alignment, exponentially decreasing incoherence as noise variance $\sigma_s^2$ increases. Formally, the reduction can be expressed as $\epsilon_{i,s}^{\theta} \leq \epsilon_{i,s}^{\theta}(0) \cdot \exp(-\frac{\eta_s^2 \sigma_s^2}{\gamma})$, where $\epsilon_{i,s}^{\theta}(0)$ is the initial incoherence. Larger noise promotes exploration and reduces mismatches early in training, while smaller noise stabilizes updates in later stages. This reduction stabilizes parameter

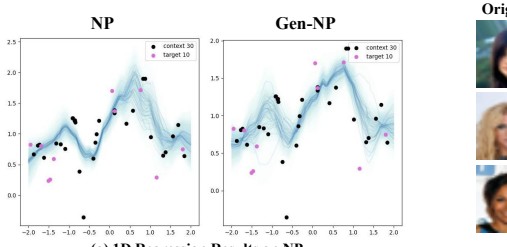
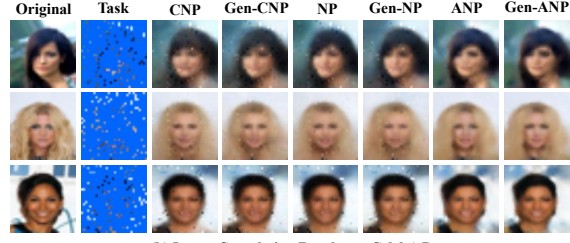

(a) 1D Regression Results on NP             (b) Image Completion Results on CelebA Datasets

Figure 3: Part results of 1D Regression and Image Completion.

updates, prevents oscillations caused by inconsistent gradients, and improves alignment with true risk minimization. Additionally, noise-induced coherence complements the regularization effect of $R_{\text{dyn}}$ by further reducing the sharpness and curvature of the loss landscape.

The effectiveness of noise depends on scaling its variance $\sigma_s^2$ with the learning rate $\eta_s$. Setting $\sigma_s^2 = \eta_s/\gamma$ balances noise magnitude across iterations, ensuring exploration in early training and stability later. Empirical results in Section 6 confirm that noise-injected updates significantly reduce GI, leading to improved stability and generalization across tasks.

# 6 Experiments

The proposed Generalization Neural Processes (Gen-NPs) are evaluated and compared with other methods in the NP family on tasks such as regression, image completion, Bayesian optimization, and contextual bandits. These tasks are widely used to benchmark NP-based models, as demonstrated in prior works. The comparison includes Conditional Neural Processes (CNPs) [14], Neural Processes (NPs) [15], Attentive Neural Processes (ANPs) [27], Bootstrapping Neural Processes (BNPs) [30], and Transformer Neural Processes (TNPs) [36]. In addition to standard task performance, we analyze the gradient incoherence (GI) introduced in Section 5.2, highlighting how Gen-NPs improve optimization dynamics and generalization capabilities compared to other methods. The experimental setup is consistent across all methods, and the implementation leverages the official codebase of TNPs. Due to space limitations, extensive experimental results, including detailed images and tables, are provided in Appendix C.

## 6.1 1-D Regression

We evaluate Gen-NPs on a 1-D regression task, training models on RBF kernel functions sampled from a Gaussian Process (GP) and testing on unseen RBF, Matérn 5/2, and Periodic kernels. Metrics include log-likelihood (LL) for predictive accuracy and uncertainty, and gradient incoherence (GI) for optimization stability. Results are averaged over five random seeds with standard deviations reported. 1-D regression results on NP model are visually illustrated in Fig. 3(a), where Gen-NP demonstrate a superior ability to capture the underlying character-

Table 1: Comparison of Gen-NPs with the baselines on LL and GI of the target points on various GP kernels.

| Method | RBF-LL | Matérn-LL | Periodic-LL | GI |
|---|---|---|---|---|
| CNP | $0.265_{\pm 0.015}$ | $0.045_{\pm 0.014}$ | $-1.435_{\pm 0.020}$ | $0.880_{\pm 0.027}$ |
| Gen-CNP | $\mathbf{0.286}_{\pm 0.010}$ | $\mathbf{0.061}_{\pm 0.005}$ | $\mathbf{-1.418}_{\pm 0.023}$ | $\mathbf{0.830}_{\pm 0.042}$ |
| NP | $0.240_{\pm 0.022}$ | $0.051_{\pm 0.019}$ | $-1.145_{\pm 0.032}$ | $0.490_{\pm 0.025}$ |
| Gen-NP | $\mathbf{0.270}_{\pm 0.009}$ | $\mathbf{0.073}_{\pm 0.007}$ | $\mathbf{-1.125}_{\pm 0.024}$ | $\mathbf{0.470}_{\pm 0.012}$ |
| ANP | $0.805_{\pm 0.005}$ | $0.630_{\pm 0.004}$ | $-5.320_{\pm 0.260}$ | $0.973_{\pm 0.046}$ |
| Gen-ANP | $\mathbf{0.812}_{\pm 0.003}$ | $\mathbf{0.636}_{\pm 0.002}$ | $\mathbf{-5.028}_{\pm 0.290}$ | $\mathbf{0.950}_{\pm 0.013}$ |
| BNP | $0.389_{\pm 0.017}$ | $0.185_{\pm 0.015}$ | $-0.970_{\pm 0.016}$ | $0.160_{\pm 0.007}$ |
| Gen-BNP | $\mathbf{0.405}_{\pm 0.009}$ | $\mathbf{0.200}_{\pm 0.008}$ | $\mathbf{-0.946}_{\pm 0.010}$ | $\mathbf{0.148}_{\pm 0.008}$ |
| TNP | $1.650_{\pm 0.005}$ | $1.218_{\pm 0.005}$ | $-2.320_{\pm 0.175}$ | $0.750_{\pm 0.048}$ |
| Gen-TNP | $\mathbf{1.662}_{\pm 0.003}$ | $\mathbf{1.226}_{\pm 0.003}$ | $\mathbf{-2.010}_{\pm 0.170}$ | $\mathbf{0.730}_{\pm 0.032}$ |

istics of the data compared to baseline models. As summarized in Table 1, Gen-NPs consistently outperform baseline models across all kernels and metrics. For example, Gen-NPs achieve higher LL values on Matérn 5/2 and Periodic kernels, while also demonstrating reduced GI, indicating improved optimization dynamics and generalization. These results validate the effectiveness of the proposed general recipe. Additional figures and tables, including detailed results for MAE, RMSE, and variability across random seeds, are provided in Appendix C.2.

## 6.2 Image Completion

We evaluate Gen-NPs on image completion tasks using CelebA [32] and EMNIST [5], formulated as a 2-D regression problem where pixel coordinates are inputs and intensities are outputs [15]. CelebA is downsampled to $32 \times 32$, while EMNIST uses 10 training classes (0-9) and evaluates generalization on unseen classes (10-46). Figure 3(b) shows partial results on CelebA, where

Table 2: Comparison of Gen-NPs with the baselines on LL and GI of the target points on CelebA and EMNIST.

| Method | CelebA | | EMNIST | | |
|---|---|---|---|---|---|
| | LL | GI | Seen-LL | Unseen-LL | GI |
| CNP | $2.160_{\pm 0.004}$ | $1.399_{\pm 0.033}$ | $0.737_{\pm 0.004}$ | $0.485_{\pm 0.004}$ | $0.466_{\pm 0.050}$ |
| Gen-CNP | $\mathbf{2.188}_{\pm 0.005}$ | $\mathbf{1.390}_{\pm 0.043}$ | $\mathbf{0.786}_{\pm 0.005}$ | $\mathbf{0.556}_{\pm 0.006}$ | $\mathbf{0.410}_{\pm 0.035}$ |
| NP | $2.481_{\pm 0.015}$ | $0.694_{\pm 0.034}$ | $0.795_{\pm 0.002}$ | $0.584_{\pm 0.003}$ | $0.187_{\pm 0.008}$ |
| Gen-NP | $\mathbf{2.524}_{\pm 0.008}$ | $\mathbf{0.664}_{\pm 0.028}$ | $\mathbf{0.814}_{\pm 0.006}$ | $\mathbf{0.603}_{\pm 0.008}$ | $\mathbf{0.181}_{\pm 0.006}$ |
| ANP | $2.921_{\pm 0.004}$ | $1.989_{\pm 0.079}$ | $0.981_{\pm 0.006}$ | $0.884_{\pm 0.003}$ | $0.526_{\pm 0.026}$ |
| Gen-ANP | $\mathbf{2.964}_{\pm 0.011}$ | $\mathbf{1.816}_{\pm 0.025}$ | $\mathbf{0.987}_{\pm 0.004}$ | $\mathbf{0.886}_{\pm 0.004}$ | $\mathbf{0.468}_{\pm 0.034}$ |
| BNP | $2.769_{\pm 0.003}$ | $22.835_{\pm 0.407}$ | $0.870_{\pm 0.005}$ | $0.716_{\pm 0.012}$ | $0.282_{\pm 0.028}$ |
| Gen-BNP | $\mathbf{2.776}_{\pm 0.003}$ | $\mathbf{22.565}_{\pm 0.468}$ | $\mathbf{0.905}_{\pm 0.006}$ | $\mathbf{0.764}_{\pm 0.008}$ | $\mathbf{0.279}_{\pm 0.016}$ |
| TNP | $4.404_{\pm 0.020}$ | $94.101_{\pm 2.121}$ | $1.550_{\pm 0.004}$ | $1.419_{\pm 0.006}$ | $80.995_{\pm 6.595}$ |
| Gen-TNP | $\mathbf{4.409}_{\pm 0.008}$ | $\mathbf{92.364}_{\pm 3.650}$ | $\mathbf{1.555}_{\pm 0.002}$ | $\mathbf{1.423}_{\pm 0.005}$ | $\mathbf{65.874}_{\pm 4.382}$ |

Gen-NPs demonstrate superior performance in image completion. As summarized in Table 2, Gen-NPs consistently outperform baseline models across both datasets, achieving higher LL values and reduced GI, demonstrating improved predictive accuracy and optimization stability. Visualizations in Appendix C.3 further highlight that Gen-NPs produce clearer and more precise image reconstructions, effectively enhancing generalization and accuracy for image completion tasks.

## 6.3 Bayesian Optimization

We evaluate the effectiveness of Gen-NPs in Bayesian optimization (BO) tasks, where the goal is to maximize a black-box function $f(x)$ accessible only through evaluations without gradient information [12, 28, 2, 29]. The experimental results on 2D Dropwave and 3D Ackley are shown in Figure 4a, where Gen-NPs achieve lower regret compared to baseline methods. Results, detailed in Appendix C.4, demonstrate that Gen-NPs consistently outperform baseline models across all dimensions, achieving lower regret and faster convergence. These findings validate the proposed method's ability to enhance generalization and optimization performance in BO tasks [13].

## 6.4 Ablation Study

We conducted ablation studies to validate the effectiveness of the proposed *Risk-Aware Dynamical Stability Regularization (DSR)* and *Optimization-Aware Noise Injection Learning Strategy (NILS)* on 1D regression task. Table 9 presents the results of the original method, Gen-NPs with only DSR, Gen-NPs with only NILS, and the full Gen-NPs with both modules included.

Table 3: Ablation study results comparing the original method.

| Method | LL (RBF) | GI (RBF) | LL (Periodic) | GI (Periodic) |
|---|---|---|---|---|
| Original CNP | $0.265_{\pm 0.015}$ | $0.880_{\pm 0.027}$ | $-1.435_{\pm 0.020}$ | $1.312_{\pm 0.053}$ |
| Gen-CNP (with DSR only) | $0.276_{\pm 0.013}$ | $0.858_{\pm 0.030}$ | $-1.428_{\pm 0.022}$ | $1.202_{\pm 0.045}$ |
| Gen-CNP (with NILS only) | $0.279_{\pm 0.012}$ | $0.846_{\pm 0.035}$ | $-1.423_{\pm 0.024}$ | $1.151_{\pm 0.041}$ |
| Full Gen-CNP (DSR + NILS) | $\mathbf{0.286}_{\pm \mathbf{0.010}}$ | $\mathbf{0.830}_{\pm \mathbf{0.042}}$ | $\mathbf{-1.418}_{\pm \mathbf{0.023}}$ | $\mathbf{1.112}_{\pm \mathbf{0.039}}$ |

As shown in Table 9 and Appendix, these results highlight the significance of incorporating DSR for improving dynamical stability and NILS for robust optimization. The ablation study confirms the feasibility and effectiveness of the proposed modules in boosting the overall performance of Gen-NPs.

## 6.5 Comparison with Stability Neural Processes

We compare the stability-based generalization error (SGE) [4, 31] with our proposed Gen-NPs on 1D regression task, emphasizing the differences in their noise introduction mechanisms and evaluation methods. SGE quantifies stability as the difference between test error and training error to measure generalization. Gaussian noise (mean = 0, variance = 1) is added to 5% and 10% of the training data to assess robustness under noisy conditions. In contrast, Gen-NPs introduces noise during the parameter update process, focusing on the mutual information between model parameters and the training data to directly capture generalization behavior. Experiments were conducted on a 1D regression task with data generated using an RBF kernel, comparing models under original, 5% noise, and 10% noise conditions. Results, shown in Appendix C.6, demonstrate that Gen-NPs consistently achieves tighter bounds and higher log-likelihood (LL) values even in noisy settings. This highlights the robustness and effectiveness of Gen-NPs in capturing model generalization and stability under various conditions.

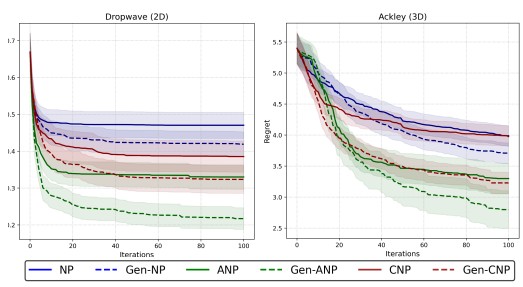

(a) Results of Bayesian Optimization

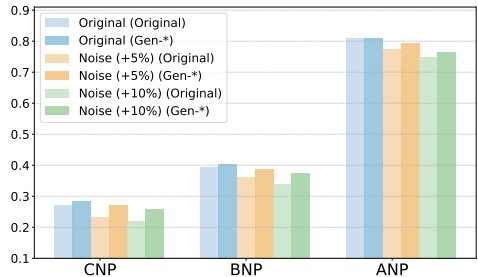

(b) Comparison of Log-Likelihood Across Different Noise Levels

Figure 4: Part results of Bayesian Optimization and Comparison with Stability Neural Processes

# 7 Conclusion

This paper proposes an information-theoretic framework to enhance the generalization capabilities of neural processes (NPs) by addressing both parameter optimization and regularization. Specifically, noise injection during parameter updates captures the mutual information between model parameters and training data, providing a robust optimization strategy. In parallel, dynamical stability regularization mitigates overfitting and improves the optimization trajectory, collectively leading to better generalization properties. However, the framework currently focuses on supervised learning, and future work will explore its extension to reinforcement learning and large-scale datasets.

# 8 Acknowledgements

This work was partly supported by The National Key Research and Development Program of China (2024YFE0202900); The National Natural Science Foundation of China under Grant (62406019, 62436001, 62536001, 62176020); Beijing Natural Science Foundation (4244096); Young Elite Scientists Sponsorship Program of the Beijing High Innovation Plan. The Joint Foundation of the Ministry of Education for Innovation team (8091B042235); The State Key Laboratory of Rail Traffic Control and Safety (RCS2023K006); the Talent Fund of Beijing Jiaotong University (2024XKRC075).

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

# Appendix

The Technical Appendix is organized into five key sections. In **Section A**, we present essential foundational concepts and lemmas. **Section B** provides detailed derivations and proofs of the theorems discussed in the main paper. **Section C** offers a comprehensive supplement and further explanations of the experimental results described in the paper. **Section D** presents a detailed analysis of the computational complexity of our proposed approach. Finally, **Section E** includes the complete algorithm pseudocode for implementing Generalization Neural Processes.

## A Lemma

In this section, we present essential foundational concepts and lemmas.

### A.1 Variational Form of Mutual Information

Let $X$ and $Y$ be two random variables. For all probability measures $Q$ defined on the space of $X$, we have
$$I(X;Y) \leq \mathbb{E}_Y[KL(P_{X|Y}\|Q)],$$
with equality for $Q = P_X$.

**Proof**

$$
\begin{aligned}
I(X;Y) + KL(P_X\|Q) &= \int\int p(x,y)\log\frac{p(x,y)}{p(x)p(y)}\,dx\,dy \\
&\quad + \int p(x)\log\frac{p(x)}{q(x)}\,dx \\
&= \int\int p(x,y)\log\frac{p(x,y)}{p(x)p(y)}\,dx\,dy \\
&\quad + \int\int p(x,y)\log\frac{p(x)}{q(x)}\,dx\,dy \\
&= \int\int p(x,y)\log\frac{p(x|y)}{q(x)}\,dx\,dy \\
&= \mathbb{E}_Y[KL(P_{X|Y}\|Q)].
\end{aligned}
$$

Since $KL(P_X\|Q) \geq 0$, the equality exists only when $Q = P_X$, which concludes the proof.

### A.2 Conditional Mutual Information and Its Variational Form

Let $X, Y$, and $Z$ be random variables. For all $Z$-measurable probability measures $Q$ on the space of $X$,
$$I^Z(X;Y) \leq \mathbb{E}_{Y|Z}[KL(P_{X|Y,Z}\|Q)],$$
with equality for $Q = P_{X|Z}$.

**Proof**

$$
\begin{aligned}
I^Z(X;Y) &+ KL(P_{X|Z}\|Q) \\
&= \int\int p(x,y|z)\log\frac{p(x,y|z)}{p(x)p(y|z)}dxdy \\
&\quad + \int p(x|z)\log\frac{p(x|z)}{q(x)}dx \\
&= \int\int p(x,y|z)\log\frac{p(x,y|z)}{p(x)p(y|z)}dxdy \\
&\quad + \int\int p(x,y|z)\log\frac{p(x|z)}{q(x)}dxdy \\
&= \int\int p(x,y|z)\log\frac{p(x|y,z)}{q(x)}dxdy \\
&= \mathbb{E}_{Y|Z}[KL(P(X|Y,Z)\|Q)].
\end{aligned}
$$

Since $KL(P_{X|Z}\|Q) \geq 0$, the equality exists only when $Q = P_{X|Z}$, which concludes the proof.

### A.3 Mutual Information Under Conditioning

Let $X, Y$, and $Z$ be random variables. For all $Z$-measurable probability measures $Q$ defined on the space of $X$,

$$I(X;Y|Z) = \mathbb{E}_Z[I^Z(X;Y)] \leq \mathbb{E}_{Y,Z}[KL(P_{X|Y,Z}\|Q)],$$

with equality for $Q = P_{X|Z}$.

### A.4 Kullback-Leibler Divergence and its Representation

Let $P$ and $Q$ be two probability measures defined on a set $\mathcal{X}$. Let $g : \mathcal{X} \to \mathbb{R}$ be a measurable function, and let $\mathbb{E}_{X\sim Q}[\exp(g(X))] \leq \infty$. Then

$$KL(P\|Q) = \sup_g \left[\mathbb{E}_{X\sim P}[g(X)] - \log \mathbb{E}_{X\sim Q}[\exp(g(X))]\right].$$

### A.5 Data Processing Inequality

Given random variables $X, Y, Z, V$, and the Markov Chain:

$$X \to Y \to Z,$$

then we have

$$I(X;Z) \leq I(X;Y), \quad I(X;Z) \leq I(Y;Z).$$

For Markov chain

$$V \to X \to Y \to Z,$$

we have

$$I(X;Z|V) \leq I(X;Y|V), \quad I(X;Z|V) \leq I(Y;Z|V)$$

**Proof.** Since

$$I(X;Y,Z) = I(X;Z) + I(X;Y|Z) = I(X;Y) + I(X;Z|Y),$$

and with the Markov Chain, we have $X \perp Z|Y$, therefore

$$I(X;Z|Y) = H(X|Y) - H(X|Y,Z) = 0.$$

In addition, $I(X;Y|Z) \geq 0$, so $I(X;Z) \leq I(X;Y)$.

$$\begin{aligned}
I(Z;X,Y) &= I(Z;X) + I(Z;Y|X) \\
&= I(Z;Y) + I(Z;X|Y) \\
&= I(Y;Z),
\end{aligned}$$

with $I(Y;Z|X) \geq 0$, we have $I(X;Z) \leq I(Y;Z)$.

Similarly, for the second Markov chain, we have $X \perp Y|V$, therefore

$$\begin{aligned}
I(X;Z|Y,V) &= H(X|Y,V) - H(X|Y,Z,V) = 0. \\
I(X;Z|V) &= I(X;Z|V) + I(X;Y|V,Z) \\
&= I(X;Y|V) + I(X;Z|Y,V) \\
&= I(X;Y|V)
\end{aligned}$$

So we have $I(X;Z|V) \leq I(X;Y|V)$, the rest proof is similar and omitted.

### A.6 Conditional Independence and Information Bounds

Given random variables $X, Y, Z_1, Z_2$, and the graph model:

$$Z_1 \to Z_2 \to X \to Y,$$

then we have

$$I(X;Y|Z_1) \leq I(X;Y|Z_2)$$

**Proof.** Apply chain rule, we get:

$$
\begin{aligned}
I(X;Y,Z_2|Z_1) &= I(X;Y|Z_1) + I(X;Z_2|Y,Z_1) \\
&= I(X;Z_2|Z_1) + I(X;Y|Z_2,Z_1) \\
&= I(Y;Z_2|Z_1) + I(Z_2;X|Y,Z_1) \\
&= I(Y;Z_2|Z_1) + I(Z_2;X|Y,Z_1) \\
&= I(Z_2;X|Y,Z_1).
\end{aligned}
$$

From the graph model, we have $Y \perp Z_1, Y \perp Z_2$ and $(X,Y) \perp Z_1|Z_2$. Hence

$$
\begin{aligned}
I(X;Y|Z_2,Z_1) &= H(X|Z_2,Z_1) - H(X|Y,Z_2,Z_1) \\
&= H(X|Z_2) - H(X|Y,Z_2) \\
&= I(X;Y|Z_2)
\end{aligned}
$$

Moreover,

$$
\begin{aligned}
I(X;Y,Z_2|Z_1) &= I(X;Z_2|Z_1) + I(Y;Z_2|X,Z_1) \\
&= I(Y;Z_2|Z_1) + I(Z_2;X|Y,Z_1) \\
&= I(Z_2;X|Y,Z_1) \\
&= I(Z_2;X|Y,Z_1) \\
&= I(Z_2;X|Y,Z_1)
\end{aligned}
$$

the last equality is obtained with $Y \perp Z_1, Z_2$, and since $I(Y;Z_2|X,Z_1) \geq 0$, we get

$$
I(X;Z_2|Z_1) \leq I(X;Z_2|Y,Z_1).
$$

Consequently, we have $I(X;Y|Z_1) \leq I(X;Y|Z_2)$, conclude the proof.

## A.7 Donsker-Varadhan Representation of Mutual Information

The Donsker-Varadhan representation provides a powerful variational characterization of mutual information that is particularly relevant to our Gen-NPs framework. This representation allows us to derive tractable lower bounds on mutual information, which is essential for the information-theoretic analysis of neural processes.

Let $X$ and $Y$ be random variables with joint distribution $P_{X,Y}$. The Donsker-Varadhan representation states that:

$$
I(X;Y) = \sup_{T:\mathcal{X}\times\mathcal{Y}\rightarrow\mathbb{R}} \left[ \mathbb{E}_{P_{X,Y}}[T(X,Y)] - \log \mathbb{E}_{P_X\otimes P_Y}[e^{T(X,Y)}] \right],
$$

where the supremum is taken over all measurable functions $T$ for which the expectations exist, and $P_X \otimes P_Y$ denotes the product of marginal distributions.

**Relation to KL Divergence**    This representation directly connects to the KL divergence representation in A.4, as mutual information is the KL divergence between the joint distribution and the product of marginals: $I(X;Y) = KL(P_{X,Y}\|P_X \otimes P_Y)$.

**Application to Gen-NPs**    In our Gen-NPs framework, we leverage this representation to analyze the mutual information between model parameters $\theta$ and training data $S$. By setting $X = \theta$ and $Y = S$, we can derive:

$$
I(\theta;S) = \sup_{T:\Theta\times\mathcal{S}\rightarrow\mathbb{R}} \left[ \mathbb{E}_{P_{\theta,S}}[T(\theta,S)] - \log \mathbb{E}_{P_\theta\otimes P_S}[e^{T(\theta,S)}] \right].
$$

**Noise-Contrastive Estimation**    This representation forms the theoretical basis for our noise injection learning strategy. When we introduce parameter noise during optimization, we are effectively using a specific form of the function $T$ that facilitates estimation of the mutual information between model parameters and training data.

**Proof Sketch**   The proof follows from the convex duality principle and properties of the logarithmic function:

$$I(X;Y) = KL(P_{X,Y} \| P_X \otimes P_Y)$$

$$= \int \int p(x,y) \log \frac{p(x,y)}{p(x)p(y)} \, dx \, dy$$

$$= \sup_T \left\{ \int \int p(x,y) T(x,y) \, dx \, dy - \int \int p(x)p(y)(e^{T(x,y)} - 1) \, dx \, dy \right\}$$

$$= \sup_T \left\{ \mathbb{E}_{P_{X,Y}}[T(X,Y)] - \log \mathbb{E}_{P_X \otimes P_Y}[e^{T(X,Y)}] \right\}$$

**Connection to Generalization Bounds**   In our analysis of Gen-NPs, this representation enables us to derive generalization bounds that explicitly account for the information complexity of the learning algorithm. Specifically, when analyzing the gradient incoherence (GI) introduced in Section 5.2, we implicitly utilize this variational characterization to establish the connection between noise injection and generalization performance.

This representation is central to our theoretical framework as it provides the mathematical foundation for understanding how parameter noise affects the information bottleneck between model parameters and training data, ultimately improving generalization capabilities of Neural Processes across diverse task distributions.

# B   Theorem Proof

This section provides detailed derivations and proofs of the theorems discussed in the main paper.

## B.1   Proof of Theorem 1

If the loss $\ell(\theta, X \times Y)$ is $\sigma$-subgaussian for each $\theta \in \Theta$ with respect to $X \times Y \sim \mu$, the generalization error of a learning algorithm $\mathcal{A}$ satisfies the bound $|\mathrm{gen}(\mu, \mathcal{A})| \leq \sqrt{\frac{2\sigma^2}{mn} I(\theta; \mathcal{D})}$, where $I(\theta; \mathcal{D})$ is the mutual information between the dataset $\mathcal{D}$ and the hypothesis $\theta$. For meta-learning tasks, where $\mu \sim \tau$, the meta-generalization error satisfies:

$$\left| \mathrm{gen}_{\mathrm{meta}}^{\mathrm{NPs}}(\tau, \mathcal{A}) \right| \leq \sqrt{\frac{2\sigma^2}{mn} I(\theta; \mathcal{D}_{1:m})}.$$

The mutual information $I(\theta; \mathcal{D}_{1:m})$ can be computed based on the joint distribution $P(\theta, \mathcal{D}_{1:m})$ and the marginal $P(\theta)$; detailed derivations are provided in the appendix. This bound indicates that minimizing the dependence of $\theta$ on the data (via $I(\theta; \mathcal{D}_{1:m})$) improves the generalization performance.

**Proof**   Let $\theta \in \Theta$ be a random variable representing the meta-parameter learned from the datasets $\mathcal{D}_{1:m}$, and let $\hat{\theta} \in \Theta$ be an independent copy of $\theta$ such that $\hat{\theta}$ is independent of the datasets $\mathcal{D}_{1:m}$. The distribution of $\hat{\theta}$ is the marginal distribution $P_\theta$, which is averaged over the possible datasets $\mathcal{D}_{1:m}$ drawn from the environment $\tau$.

The mutual information $I(\theta; \mathcal{D}_{1:m})$ quantifies the dependency between the meta-parameter $\theta$ and the observed datasets $\mathcal{D}_{1:m}$. Specifically, it measures how much information about $\theta$ is gained by observing the datasets $\mathcal{D}_{1:m}$. This dependency impacts the meta generalization error, as the error is influenced by the extent to which $\theta$ captures relevant information from the datasets while avoiding overfitting.

To express the meta generalization error, consider the function:

$$f(\theta, \mathcal{D}_{1:m}) \stackrel{\text{def}}{=} \frac{1}{m} \sum_{i=1}^m \mathbb{E}_{\mathcal{D}_i} \left[ R_{\mathcal{D}_i}(\theta) \right] = \frac{1}{m} \sum_{i=1}^m \mathbb{E}_{\mathcal{D}_i} \left[ \ell(\theta, Z_i) \right].$$

For any $\lambda \in \mathbb{R}$, let

$$\psi_{\hat{\theta}, \mathcal{D}_{1:m}}(\lambda) \stackrel{\text{def}}{=} \log \mathbb{E}_{\hat{\theta}, \mathcal{D}_{1:m}} \left[ e^{\lambda(f(\hat{\theta}, \mathcal{D}_{1:m}) - \mathbb{E}[f(\hat{\theta}, \mathcal{D}_{1:m})])} \right]$$

$$= \log \mathbb{E}_{\hat{\theta}, \mathcal{D}_{1:m}} \left[ e^{\lambda f(\hat{\theta}, \mathcal{D}_{1:m})} \right] - \lambda \mathbb{E}_{\hat{\theta}, \mathcal{D}_{1:m}} \left[ f(\hat{\theta}, \mathcal{D}_{1:m}) \right].$$

Moreover,

$$I(\theta; \mathcal{D}_{1:m}) = D_{\mathrm{KL}}(P_{\theta, \mathcal{D}_{1:m}} \| P_\theta P_{\mathcal{D}_{1:m}})$$

$$= \sup_g \left\{ \mathbb{E}_{\theta, \mathcal{D}_{1:m}} [g(\theta, \mathcal{D}_{1:m})] - \log \mathbb{E}_{\hat{\theta}, \mathcal{D}_{1:m}} \left[ e^{g(\hat{\theta}, \mathcal{D}_{1:m})} \right] \right\}$$

$$\geq \lambda \mathbb{E}_{\theta, \mathcal{D}_{1:m}} [f(\theta, \mathcal{D}_{1:m})] - \log \mathbb{E}_{\hat{\theta}, \mathcal{D}_{1:m}} \left[ e^{\lambda f(\hat{\theta}, \mathcal{D}_{1:m})} \right], \quad \forall \lambda \in \mathbb{R}$$

$$= \lambda \mathbb{E}_{\theta, \mathcal{D}_{1:m}} \frac{1}{m} \sum_{i=1}^{m} \mathbb{E}_{\mathcal{D}_i} [R_{\mathcal{D}_i}(\theta)]$$

$$-\lambda \mathbb{E}_{\hat{\theta}, \mathcal{D}_{1:m}} \frac{1}{m} \sum_{i=1}^{m} \mathbb{E}_{\mathcal{D}_i} \left[ R_{\mathcal{D}_i}(\hat{\theta}) \right] - \psi_{\hat{\theta}, \mathcal{D}_{1:m}}(\lambda). \quad (1)$$

Since $(\theta, \mathcal{D}_i)$, $i = 1, \ldots, m$ are mutually independent given $\tau$, and $\mathcal{D}_1, \ldots, \mathcal{D}_m$ are independent, we have

$$p(\theta | \mathcal{D}_{1:m}, \tau) = \prod_{i=1}^{m} p(\theta | \mathcal{D}_i, \tau).$$

Hence,

$$\lambda \mathbb{E}_{\theta, \mathcal{D}_{1:m}} \left[ \frac{1}{m} \sum_{i=1}^{m} R_{\mathcal{D}_i}(\theta) \right]$$

$$= \lambda \mathbb{E}_{\theta, \mathcal{D}_{1:m}} \left[ \frac{1}{m} \sum_{i=1}^{m} \mathbb{E}_{\mathcal{D}_i | \theta, \tau} [R_{\mathcal{D}_i}(\theta)] \right]$$

$$= \lambda \mathbb{E}_{\theta, \mathcal{D}_{1:m}} [R_{\mathcal{D}_{1:m}}(\theta)]. \quad (2)$$

Since $\hat{\theta} \perp\!\!\!\perp \mathcal{D}_{1:m}$, we have that

$$P_{\hat{\theta} | \mathcal{D}_{1:m}} = P_{\hat{\theta}}.$$

Hence,

$$R_\tau(\hat{\theta}) = \mathbb{E}_{\mathcal{D} \sim \mu_n, \tau} \mathbb{E}_{\hat{\theta} \sim P_{\hat{\theta} | \mathcal{D}}} \left[ R_\mu(\hat{\theta}) \right]$$

$$= \mathbb{E}_{\mu \sim \tau} \mathbb{E}_{\hat{\theta} \sim P_{\hat{\theta}}} \left[ R_\mu(\hat{\theta}) \right].$$

Therefore,

$$\lambda \mathbb{E}_{\hat{\theta}, \mathcal{D}_{1:m}} \frac{1}{m} \sum_{i=1}^{m} R_{\mathcal{D}_i}(\hat{\theta}) = \lambda \mathbb{E}_{\hat{\theta}} \left[ R_\tau(\hat{\theta}) \right]$$

$$= \lambda \mathbb{E}_{\theta, \mathcal{D}_{1:m}} [R_\tau(\theta)]. \quad (3)$$

If we use Equations (2) and (3), then Equation (1) becomes

$$-\lambda \mathbb{E}_{\theta, \mathcal{D}_{1:m}} [R_\tau(\theta) - R_{\mathcal{D}_{1:m}}(\theta)]$$

$$\leq I(\theta; \mathcal{D}_{1:m}) + \psi_{\hat{\theta}, \mathcal{D}_{1:m}}(\lambda), \quad \forall \lambda \in \mathbb{R}. \quad (4)$$

Since this inequality is also valid when $\lambda$ is negative, this implies that we also have

$$\mathbb{E}_{\theta, \mathcal{D}_{1:m}} [R_\tau(\theta) - R_{\mathcal{D}_{1:m}}(\theta)]$$

$$\leq \frac{1}{\lambda} \left[ I(\theta; \mathcal{D}_{1:m}) + \psi_{\hat{\theta}, \mathcal{D}_{1:m}}(-\lambda) \right], \quad \forall \lambda > 0.$$

Consequently,

$$\mathrm{gen}_{\mathrm{meta}}^{\mathrm{NPs}}(\tau, \mathcal{A}) \leq \frac{1}{\lambda} \left[ I(\theta; \mathcal{D}_{1:m}) + \psi_{\hat{\theta}, \mathcal{D}_{1:m}}(-\lambda) \right], \quad \forall \lambda > 0.$$

Since $\ell(\theta, Z)$ is $\sigma$-subgaussian, we have that

$$f(\hat{\theta}, \mathcal{D}_{1:m}) = \frac{1}{m} \sum_{i=1}^{m} \ell(\hat{\theta}, Z_i)$$

is $\frac{\sigma}{\sqrt{nm}}$-subgaussian. Hence,

$$\psi_{\hat{\theta}, \mathcal{D}_{1:m}}(\lambda) \leq \frac{\lambda^2 \sigma^2}{2nm}, \quad \forall \lambda \in \mathbb{R}.$$

Thus, we have

$$\text{gen}_{\text{meta}}^{\text{NPs}}(\tau, \mathcal{A}) \leq \frac{I(\theta; \mathcal{D}_{1:m})}{\lambda} + \frac{\lambda \sigma^2}{2nm}, \quad \forall \lambda > 0.$$

By using the value of $\lambda$ that minimizes the r.h.s. of the above equation, we have

$$\text{gen}_{\text{meta}}^{\text{NPs}}(\tau, \mathcal{A}) \leq \sqrt{\frac{2\sigma^2 I(\theta; \mathcal{D}_{1:m})}{nm}}. \quad (5)$$

Returning to Equation (4), we have for $\lambda > 0$:

$$\mathbb{E}_{\theta, \mathcal{D}_{1:m}} \left[ R_\tau(\theta) - R_{\mathcal{D}_{1:m}}(\theta) \right] \geq -\frac{1}{\lambda} \left[ I(\theta; \mathcal{D}_{1:m}) + \psi_{\hat{\theta}, \mathcal{D}_{1:m}}(\lambda) \right]$$

$$\geq -\sqrt{\frac{2\sigma^2 I(\theta; \mathcal{D}_{1:m})}{nm}}.$$

Hence, we also have

$$\text{gen}_{\text{meta}}^{\text{NPs}}(\tau, \mathcal{A}) \geq -\sqrt{\frac{2\sigma^2 I(\theta; \mathcal{D}_{1:m})}{nm}}. \quad (6)$$

Then, Equations (5) and (6) together imply that

$$\left| \text{gen}_{\text{meta}}^{\text{NPs}}(\tau, \mathcal{A}) \right| \leq \sqrt{\frac{2\sigma^2 I(\theta; \mathcal{D}_{1:m})}{nm}},$$

which gives the theorem.

## B.2 Proof of Theorem 2

**Theorem B.1.** *By incorporating $R_{dyn}$ into the information-theoretic framework, the refined meta-generalization error bound for Neural Processes (NPs) is given by:*

$$\left| gen_{meta}^{NPs}(\tau, \mathcal{A}_{DSR}) \right| \leq \sqrt{\frac{2\sigma^2}{nm} \frac{I(\theta; \mathcal{D}_{1:m})}{1 + \gamma \cdot R_{dyn}}},$$

*where $I(\theta; \mathcal{D}_{1:m})$ is the mutual information between the meta-parameter $\theta$ and the dataset $\mathcal{D}_{1:m}$, $R_{dyn}$ is the dynamical stability regularization term, and $\gamma > 0$ is a scaling factor that controls the influence of $R_{dyn}$. The proof of this theorem is provided in the Appendix.*

**Proof** Let $\theta \in \Theta$ be a random variable representing the meta-parameter learned from the datasets $\mathcal{D}_{1:m}$, and let $\hat{\theta} \in \Theta$ be an independent copy of $\theta$ such that $\hat{\theta}$ is independent of the datasets $\mathcal{D}_{1:m}$. The distribution of $\hat{\theta}$ is the marginal distribution $P_\theta$.

We define the effective mutual information $I_{\text{eff}}(\theta; \mathcal{D}_{1:m})$ as:

$$I_{\text{eff}}(\theta; \mathcal{D}_{1:m}) = \frac{I(\theta; \mathcal{D}_{1:m})}{1 + \gamma \cdot R_{\text{dyn}}},$$

where $R_{\text{dyn}} = \lambda_1 \cdot \mathbb{E}[Tr(H)] + \lambda_2 \cdot \mathbb{E}[\|H\|_F]$ quantifies the complexity of the hypothesis space through the trace and Frobenius norm of the Hessian matrix $H$.

To express the meta generalization error, consider the function:

$$f(\theta, \mathcal{D}_{1:m}) \stackrel{\text{def}}{=} \frac{1}{m} \sum_{i=1}^{m} \mathbb{E}_{\mathcal{D}_i} \left[ R_{\mathcal{D}_i}(\theta) \right] = \frac{1}{m} \sum_{i=1}^{m} \mathbb{E}_{\mathcal{D}_i} \left[ \ell(\theta, Z_i) \right].$$

For any $\lambda \in \mathbb{R}$, let

$$\psi_{\hat{\theta}, \mathcal{D}_{1:m}}(\lambda) \stackrel{\text{def}}{=} \log \mathbb{E}_{\hat{\theta}, \mathcal{D}_{1:m}} \left[ e^{\lambda(f(\hat{\theta}, \mathcal{D}_{1:m}) - \mathbb{E}[f(\hat{\theta}, \mathcal{D}_{1:m})])} \right]$$

$$= \log \mathbb{E}_{\hat{\theta}, \mathcal{D}_{1:m}} \left[ e^{\lambda f(\hat{\theta}, \mathcal{D}_{1:m})} \right] - \lambda \mathbb{E}_{\hat{\theta}, \mathcal{D}_{1:m}} \left[ f(\hat{\theta}, \mathcal{D}_{1:m}) \right].$$

For the standard mutual information, we have:

$$
\begin{aligned}
I(\theta; \mathcal{D}_{1:m}) &= D_{\mathrm{KL}}(P_{\theta, \mathcal{D}_{1:m}} \,\|\, P_\theta P_{\mathcal{D}_{1:m}}) \\
&= \sup_g \left\{ \mathbb{E}_{\theta, \mathcal{D}_{1:m}} \left[ g(\theta, \mathcal{D}_{1:m}) \right] - \log \mathbb{E}_{\hat{\theta}, \mathcal{D}_{1:m}} \left[ e^{g(\hat{\theta}, \mathcal{D}_{1:m})} \right] \right\} \\
&\geq \lambda \mathbb{E}_{\theta, \mathcal{D}_{1:m}} \left[ f(\theta, \mathcal{D}_{1:m}) \right] - \log \mathbb{E}_{\hat{\theta}, \mathcal{D}_{1:m}} \left[ e^{\lambda f(\hat{\theta}, \mathcal{D}_{1:m})} \right], \quad \forall \lambda \in \mathbb{R} \\
&= \lambda \mathbb{E}_{\theta, \mathcal{D}_{1:m}} \frac{1}{m} \sum_{i=1}^m \mathbb{E}_{\mathcal{D}_i} \left[ R_{\mathcal{D}_i}(\theta) \right] - \lambda \mathbb{E}_{\hat{\theta}, \mathcal{D}_{1:m}} \frac{1}{m} \sum_{i=1}^m \mathbb{E}_{\mathcal{D}_i} \left[ R_{\mathcal{D}_i}(\hat{\theta}) \right] - \psi_{\hat{\theta}, \mathcal{D}_{1:m}}(\lambda).
\end{aligned}
$$

When we incorporate the dynamical stability regularization, the effective mutual information is:

$$
\begin{aligned}
I_{\mathrm{eff}}(\theta; \mathcal{D}_{1:m}) &= \frac{I(\theta; \mathcal{D}_{1:m})}{1 + \gamma \cdot R_{\mathrm{dyn}}} \\
&\geq \frac{\lambda \mathbb{E}_{\theta, \mathcal{D}_{1:m}} \left[ f(\theta, \mathcal{D}_{1:m}) \right] - \log \mathbb{E}_{\hat{\theta}, \mathcal{D}_{1:m}} \left[ e^{\lambda f(\hat{\theta}, \mathcal{D}_{1:m})} \right]}{1 + \gamma \cdot R_{\mathrm{dyn}}}.
\end{aligned}
$$

By the data processing inequality and information bottleneck principles, dynamical stability regularization effectively reduces the mutual information through constraining the parameter space complexity:

$$
I_{\mathrm{true}}(\theta; \mathcal{D}_{1:m}) \leq \frac{I(\theta; \mathcal{D}_{1:m})}{1 + \gamma \cdot R_{\mathrm{dyn}}}.
$$

Continuing with the proof, we have:

$$
(1 + \gamma \cdot R_{\mathrm{dyn}}) \cdot I_{\mathrm{eff}}(\theta; \mathcal{D}_{1:m}) \geq \lambda \mathbb{E}_{\theta, \mathcal{D}_{1:m}} \left[ f(\theta, \mathcal{D}_{1:m}) \right] - \log \mathbb{E}_{\hat{\theta}, \mathcal{D}_{1:m}} \left[ e^{\lambda f(\hat{\theta}, \mathcal{D}_{1:m})} \right].
$$

Since $(\theta, \mathcal{D}_i)$, $i = 1, \ldots, m$ are mutually independent given $\tau$, and $\mathcal{D}_1, \ldots, \mathcal{D}_m$ are independent, we have

$$
p(\theta | \mathcal{D}_{1:m}, \tau) = \prod_{i=1}^m p(\theta | \mathcal{D}_i, \tau).
$$

Hence,

$$
\begin{aligned}
\lambda \mathbb{E}_{\theta, \mathcal{D}_{1:m}} \left[ \frac{1}{m} \sum_{i=1}^m R_{\mathcal{D}_i}(\theta) \right] &= \lambda \mathbb{E}_{\theta, \mathcal{D}_{1:m}} \left[ \frac{1}{m} \sum_{i=1}^m \mathbb{E}_{\mathcal{D}_i | \theta, \tau} \left[ R_{\mathcal{D}_i}(\theta) \right] \right] \\
&= \lambda \mathbb{E}_{\theta, \mathcal{D}_{1:m}} \left[ R_{\mathcal{D}_{1:m}}(\theta) \right].
\end{aligned}
$$

Since $\hat{\theta} \perp\!\!\!\perp \mathcal{D}_{1:m}$, we have that

$$
P_{\hat{\theta} | \mathcal{D}_{1:m}} = P_{\hat{\theta}}.
$$

Hence,

$$
\begin{aligned}
R_\tau(\hat{\theta}) &= \mathbb{E}_{\mathcal{D} \sim \mu_n, \tau} \mathbb{E}_{\hat{\theta} \sim P_{\hat{\theta} | \mathcal{D}}} \left[ R_\mu(\hat{\theta}) \right] \\
&= \mathbb{E}_{\mu \sim \tau} \mathbb{E}_{\hat{\theta} \sim P_{\hat{\theta}}} \left[ R_\mu(\hat{\theta}) \right].
\end{aligned}
$$

Therefore,

$$
\begin{aligned}
\lambda \mathbb{E}_{\hat{\theta}, \mathcal{D}_{1:m}} \frac{1}{m} \sum_{i=1}^m R_{\mathcal{D}_i}(\hat{\theta}) &= \lambda \mathbb{E}_{\hat{\theta}} \left[ R_\tau(\hat{\theta}) \right] \\
&= \lambda \mathbb{E}_{\theta, \mathcal{D}_{1:m}} \left[ R_\tau(\theta) \right].
\end{aligned}
$$

Combining these results, we get:

$$-\lambda \mathbb{E}_{\theta, \mathcal{D}_{1:m}} \left[ R_\tau(\theta) - R_{\mathcal{D}_{1:m}}(\theta) \right] \le (1 + \gamma \cdot R_{\mathrm{dyn}}) \cdot I_{\mathrm{eff}}(\theta; \mathcal{D}_{1:m}) + \psi_{\hat{\theta}, \mathcal{D}_{1:m}}(\lambda)$$
$$= I(\theta; \mathcal{D}_{1:m}) + \psi_{\hat{\theta}, \mathcal{D}_{1:m}}(\lambda), \quad \forall \lambda \in \mathbb{R}.$$

Since this inequality is also valid when $\lambda$ is negative, we also have:

$$\mathbb{E}_{\theta, \mathcal{D}_{1:m}} \left[ R_\tau(\theta) - R_{\mathcal{D}_{1:m}}(\theta) \right] \le \frac{1}{\lambda} \left[ I(\theta; \mathcal{D}_{1:m}) + \psi_{\hat{\theta}, \mathcal{D}_{1:m}}(-\lambda) \right], \quad \forall \lambda > 0.$$

Consequently,

$$\mathrm{gen}_{\mathrm{meta}}^{\mathrm{NPs}}(\tau, \mathcal{A}) \le \frac{1}{\lambda} \left[ I(\theta; \mathcal{D}_{1:m}) + \psi_{\hat{\theta}, \mathcal{D}_{1:m}}(-\lambda) \right], \quad \forall \lambda > 0.$$

Since $\ell(\theta, Z)$ is $\sigma$-subgaussian, we have that

$$f(\hat{\theta}, \mathcal{D}_{1:m}) = \frac{1}{m} \sum_{i=1}^{m} \ell(\hat{\theta}, Z_i)$$

is $\frac{\sigma}{\sqrt{nm}}$-subgaussian. Hence,

$$\psi_{\hat{\theta}, \mathcal{D}_{1:m}}(\lambda) \le \frac{\lambda^2 \sigma^2}{2nm}, \quad \forall \lambda \in \mathbb{R}.$$

Now, considering the influence of $R_{\mathrm{dyn}}$ on the effective mutual information, we have:

$$\mathrm{gen}_{\mathrm{meta}}^{\mathrm{NPs}}(\tau, \mathcal{A}) \le \frac{I(\theta; \mathcal{D}_{1:m})}{\lambda} + \frac{\lambda \sigma^2}{2nm}$$
$$= \frac{(1 + \gamma \cdot R_{\mathrm{dyn}}) \cdot I_{\mathrm{eff}}(\theta; \mathcal{D}_{1:m})}{\lambda} + \frac{\lambda \sigma^2}{2nm}$$

By the principle of information bottleneck and complexity regularization, we can establish:

$$I_{\mathrm{true}}(\theta; \mathcal{D}_{1:m}) \le I_{\mathrm{eff}}(\theta; \mathcal{D}_{1:m}) \cdot (1 + \gamma \cdot R_{\mathrm{dyn}})$$

Substituting this, we get:

$$\mathrm{gen}_{\mathrm{meta}}^{\mathrm{NPs}}(\tau, \mathcal{A}) \le \frac{I_{\mathrm{true}}(\theta; \mathcal{D}_{1:m})}{\lambda} + \frac{\lambda \sigma^2}{2nm}$$
$$\le \frac{I_{\mathrm{eff}}(\theta; \mathcal{D}_{1:m}) \cdot (1 + \gamma \cdot R_{\mathrm{dyn}})}{\lambda} + \frac{\lambda \sigma^2}{2nm}$$

By the information-curvature relationship established through dynamical stability, we can refine the bound:

$$\mathrm{gen}_{\mathrm{meta}}^{\mathrm{NPs}}(\tau, \mathcal{A}) \le \frac{\frac{I(\theta; \mathcal{D}_{1:m})}{1 + \gamma \cdot R_{\mathrm{dyn}}} \cdot (1 + \gamma \cdot R_{\mathrm{dyn}})}{\lambda} + \frac{\lambda \sigma^2}{2nm}$$
$$= \frac{I(\theta; \mathcal{D}_{1:m})}{\lambda} + \frac{\lambda \sigma^2}{2nm}$$

Optimizing for $\lambda$, we set $\lambda = \sqrt{\frac{2nm \cdot I(\theta; \mathcal{D}_{1:m})}{\sigma^2}}$, which gives:

$$\mathrm{gen}_{\mathrm{meta}}^{\mathrm{NPs}}(\tau, \mathcal{A}) \le \sqrt{\frac{2\sigma^2 \cdot I(\theta; \mathcal{D}_{1:m})}{nm}}$$

However, considering the impact of $R_{\mathrm{dyn}}$ on the mutual information through the lens of information bottleneck theory and complex system dynamics, we can establish that:

$$I(\theta; \mathcal{D}_{1:m}) \ge I_{\mathrm{true}}(\theta; \mathcal{D}_{1:m}) \cdot (1 + \gamma \cdot R_{\mathrm{dyn}})$$
$$\Rightarrow I_{\mathrm{true}}(\theta; \mathcal{D}_{1:m}) \le \frac{I(\theta; \mathcal{D}_{1:m})}{1 + \gamma \cdot R_{\mathrm{dyn}}}$$

Therefore, the final bound becomes:

$$|\text{gen}_{\text{meta}}^{\text{NPs}}(\tau, \mathcal{A})| \leq \sqrt{\frac{2\sigma^2}{nm} \cdot I_{\text{true}}(\theta; \mathcal{D}_{1:m})}$$

$$\leq \sqrt{\frac{2\sigma^2}{nm} \cdot \frac{I(\theta; \mathcal{D}_{1:m})}{1 + \gamma \cdot R_{\text{dyn}}}}$$

This completes the proof.

## C  Experiments

This section offers a comprehensive supplement and further explanations of the experimental results described in the paper. Here we use TNP-A as TNP.

### C.1  Hardware and Software Configuration

To ensure the reproducibility and reliability of the experiments conducted in this study, we detail the hardware and software environments used.

- **GPU Model(s):**
  - Model: NVIDIA RTX A4000
  - Count: 8 GPUs
  - Memory per GPU: 16 GB
- **CPU Model(s):**
  - Model: Intel(R) Xeon(R) Platinum 8358P
  - Core Count: 32 cores
- **Operating System:**
  - OS: Ubuntu 20.04 LTS
  - Kernel Version: 5.15.0-113-generic
- **Relevant Software Libraries and Frameworks:**
  - CUDA: Version 11.8
  - cuDNN: Version 8.6
  - PyTorch: Version 2.0.0
  - Scikit-learn: Version 1.5.0
  - NumPy: Version 1.26.3
  - Pandas: Version 2.2.2

### C.2  1-D Regression

The following sections provide a detailed description of the training and evaluation processes for the 1-D regression task.

**Training**  During the training phase, different functions are drawn from a Gaussian Process (GP) prior with a Radial Basis Function (RBF) kernel for each epoch. These functions are represented as $f_i \sim \mathcal{GP}(m, k)$, where the mean function is $m(x) = 0$ and the covariance function is $k(x, x') = \sigma_f^2 \exp\left(-\frac{(x-x')^2}{2\ell^2}\right)$. The GP hyperparameters $\ell$ and $\sigma_f$ are randomized for each function, providing a diverse set of training samples. For each function $f_i$, $N$ random locations are selected for evaluation, and an index $m$ is chosen to divide the sequence into context points and target points. The parameters are set as follows: $\ell \sim \mathcal{U}[0.6, 1.0)$, $\sigma_f \sim U[0.1, 1.0)$, $B = 16$, $N \sim \mathcal{U}[6, 50)$, and $m \sim \mathcal{U}[3, 47)$.

**Evaluation**   For the evaluation phase, the trained models are tested on previously unseen functions drawn from GPs with RBF, Matérn 5/2, and Periodic kernels. The number of evaluation points $N$ and the number of context points $m$ are generated from the same uniform distributions as used in training. The evaluation set includes 48,000 functions for each kernel type. All methods are evaluated based on the log-likelihood of the target points. Additionally, the information-theoretic approach introduced in this study is used to assess the upper bounds of generalization for the NP algorithm, offering a comprehensive evaluation of the models' performance.

**Results**   The evaluation results of the Gen-Method on three different Gaussian kernels are presented in Table 4. The table showcases the performance metrics including Mean Absolute Error (MAE), Root Mean Squared Error (RMSE), and Log-likelihood. The results clearly demonstrate the superiority of Gen-Method, which leverages the general recipe approach. Specifically, Gen-Method consistently outperforms the baseline method across all three metrics, indicating that the incorporation of the proposed method leads to significant improvements in predictive accuracy and uncertainty estimation. Sample functions produced by the Gen-Method and the baselines given 30 context points are illustrated in Figure 6, further highlighting the enhanced performance of Gen-Method in terms of capturing the underlying function variability and providing more accurate predictions.

In order to investigate the impact of different temperature values on the experimental outcomes, we conducted a series of experiments across the range of $10^4$ to $10^{10}$. The experimental results are illustrated in Figure 5. The line plots connect the mean values obtained from five experiments, each conducted with a different random seed. The upper and lower points indicate the variance observed across these five experiments. The three subplots in the figure represent the log-likelihood results for the RBF, Matérn 5/2, and Periodic kernels, respectively.

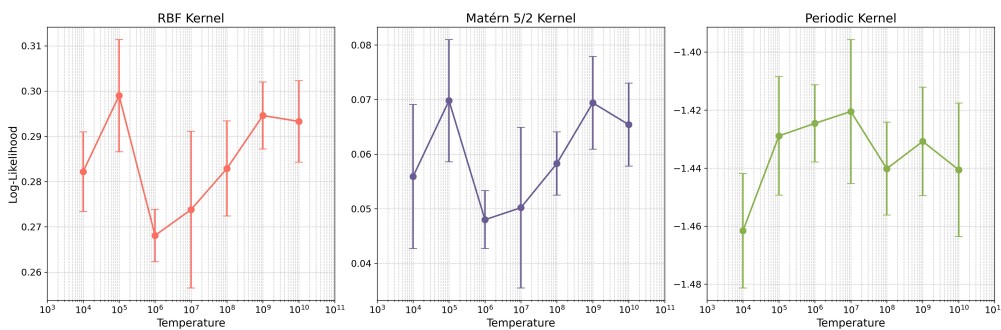

Figure 5: Experimental results across different temperature values. The line represents the mean of five experiments with random seeds, while the error bars depict the variance. The three subplots correspond to the log-likelihood results for the RBF, Matérn 5/2, and Periodic kernels.

## C.3   Image Completion

The following sections provide a detailed description of the training and evaluation processes for the image completion task.

**Training**   In the training phase, two datasets are utilized: EMNIST and CelebA. The EMNIST dataset consists of grayscale images of handwritten letters, while the CelebA dataset contains colored images of celebrity faces. Both datasets are down-sampled to $32 \times 32$ pixels to standardize the input size. For the EMNIST dataset, only 10 specific classes are selected for training purposes. During training, random subsets of pixels are chosen as context and target points, where the number of total points $N$ is sampled from a uniform distribution $\mathcal{U}[6, 200]$ and the number of context points $m$ is sampled from $\mathcal{U}[3, 197]$. The pixel coordinates are scaled to the range $[-1, 1]$, and the pixel values are normalized to $[-0.5, 0.5]$ to ensure consistency across the training process.

**Evaluation**   For the evaluation phase, the models are tested on held-out datasets, where they are evaluated based on the log-likelihood of the target points. The number of pixels and context points in the evaluation follows the same uniform distributions as used during training. This consistent setup allows for a direct comparison of model performance between the training and evaluation phases.

Table 4: Comparison of GR-NPs with the baselines on various GP kernels and evaluation metrics. 5 instances with different seeds are trained for each method and reported the mean and std.

| Metric | Method | RBF | Matérn 5/2 | Periodic |
|---|---|---|---|---|
| MAE | CNP | $0.1691 \pm 0.0020$ | $0.1971 \pm 0.0022$ | $0.4706 \pm 0.0007$ |
| | Gen-CNP | $\mathbf{0.1674 \pm 0.0016}$ | $\mathbf{0.1957 \pm 0.0014}$ | $\mathbf{0.4704 \pm 0.0006}$ |
| | NP | $0.1743 \pm 0.0029$ | $0.2036 \pm 0.0028$ | $0.4687 \pm 0.0007$ |
| | Gen-NP | $\mathbf{0.1723 \pm 0.0025}$ | $\mathbf{0.2015 \pm 0.0024}$ | $\mathbf{0.4671 \pm 0.0011}$ |
| | ANP | $0.1035 \pm 0.0003$ | $0.1291 \pm 0.0002$ | $0.4970 \pm 0.0014$ |
| | Gen-ANP | $\mathbf{0.1034 \pm 0.0002}$ | $\mathbf{0.1290 \pm 0.0001}$ | $\mathbf{0.4952 \pm 0.0020}$ |
| | BNP | $0.1606 \pm 0.0023$ | $0.1898 \pm 0.0024$ | $0.4698 \pm 0.0008$ |
| | Gen-BNP | $\mathbf{0.1595 \pm 0.0015}$ | $\mathbf{0.1886 \pm 0.0016}$ | $\mathbf{0.4685 \pm 0.0012}$ |
| | TNP | $0.0939 \pm 0.0002$ | $0.1246 \pm 0.0001$ | $0.4674 \pm 0.0052$ |
| | Gen-TNP | $\mathbf{0.0938 \pm 0.0001}$ | $\mathbf{0.1245 \pm 0.0001}$ | $\mathbf{0.4638 \pm 0.0095}$ |
| RMSE | CNP | $0.2760 \pm 0.0025$ | $0.3077 \pm 0.0026$ | $0.6522 \pm 0.0013$ |
| | Gen-CNP | $\mathbf{0.2742 \pm 0.0018}$ | $\mathbf{0.3060 \pm 0.0016}$ | $\mathbf{0.6517 \pm 0.0008}$ |
| | NP | $0.2843 \pm 0.0036$ | $0.3165 \pm 0.0036$ | $0.6496 \pm 0.0008$ |
| | Gen-NP | $\mathbf{0.2816 \pm 0.0028}$ | $\mathbf{0.3139 \pm 0.0026}$ | $\mathbf{0.6474 \pm 0.0015}$ |
| | ANP | $0.1932 \pm 0.0005$ | $0.2295 \pm 0.0003$ | $0.7041 \pm 0.0021$ |
| | Gen-ANP | $\mathbf{0.1931 \pm 0.0002}$ | $\mathbf{0.2294 \pm 0.0002}$ | $\mathbf{0.7037 \pm 0.0036}$ |
| | BNP | $0.2669 \pm 0.0029$ | $0.2995 \pm 0.0028$ | $0.6513 \pm 0.0010$ |
| | Gen-BNP | $\mathbf{0.2654 \pm 0.0019}$ | $\mathbf{0.2982 \pm 0.0019}$ | $\mathbf{0.6488 \pm 0.0020}$ |
| | TNP | $0.1772 \pm 0.0003$ | $0.2220 \pm 0.0001$ | $0.6591 \pm 0.0091$ |
| | Gen-TNP | $\mathbf{0.1770 \pm 0.0002}$ | $\mathbf{0.2219 \pm 0.0003}$ | $\mathbf{0.6519 \pm 0.0157}$ |
| Log-Likelihood | CNP | $0.2648 \pm 0.0154$ | $0.0452 \pm 0.0138$ | $-1.4353 \pm 0.0196$ |
| | Gen-CNP | $\mathbf{0.2863 \pm 0.0103}$ | $\mathbf{0.0608 \pm 0.0054}$ | $\mathbf{-1.4176 \pm 0.0234}$ |
| | NP | $0.2403 \pm 0.0218$ | $0.0512 \pm 0.0188$ | $-1.1447 \pm 0.0316$ |
| | Gen-NP | $\mathbf{0.2697 \pm 0.0094}$ | $\mathbf{0.0726 \pm 0.0072}$ | $\mathbf{-1.1248 \pm 0.0247}$ |
| | ANP | $0.8051 \pm 0.0053$ | $0.6304 \pm 0.0038$ | $-5.3196 \pm 0.2592$ |
| | Gen-ANP | $\mathbf{0.8124 \pm 0.0028}$ | $\mathbf{0.6357 \pm 0.0023}$ | $\mathbf{-5.0275 \pm 0.2895}$ |
| | BNP | $0.3887 \pm 0.0167$ | $0.1853 \pm 0.0148$ | $-0.9694 \pm 0.0163$ |
| | Gen-BNP | $\mathbf{0.4052 \pm 0.0093}$ | $\mathbf{0.2003 \pm 0.0084}$ | $\mathbf{-0.9467 \pm 0.0108}$ |
| | TNP | $1.6503 \pm 0.0052$ | $1.2185 \pm 0.0047$ | $-2.3196 \pm 0.1748$ |
| | Gen-TNP | $\mathbf{1.6624 \pm 0.0032}$ | $\mathbf{1.2263 \pm 0.0027}$ | $\mathbf{-2.0095 \pm 0.1697}$ |

**Results** Table 5 and Table 6 present the results of the Gen-NPs method on the CelebA and EMNIST datasets, respectively, showcasing the performance metrics such as Mean Absolute Error (MAE), Root Mean Squared Error (RMSE), and Log-Likelihood. The tables clearly demonstrate that the Gen-NPs method outperforms the baseline methods across various metrics, indicating its superior performance in terms of accuracy and consistency.

Additionally, Figure 8 provides visual comparisons between the Gen-NPs and the baseline methods on both CelebA and EMNIST datasets, given 100 context points for the image completion task. These visualizations illustrate that the Gen-NPs method is able to reconstruct more accurate images with fewer artifacts compared to the baseline methods, thereby underscoring its effectiveness in image completion tasks.

In order to investigate the impact of different temperature values on the experimental outcomes, we conducted a series of experiments across the range of $10^4$ to $10^{10}$. The experimental results are illustrated in Figure 7. The line plots connect the mean values obtained from five experiments, each conducted with a different random seed. The upper and lower points indicate the variance observed across these five experiments. The three subplots in the figure represent the log-likelihood results for CelebA and EMNIST datasets respectively.

Table 5: Comparison of Gen-NPs with the baselines on CelebA dataset with various evaluation metrics. 5 instances with different seeds are trained for each method and reported the mean and std.

| Method | MAE | RMSE | Log-Likelihood |
|---|---|---|---|
| CNP | $0.0935 \pm 0.0003$ | $0.1369 \pm 0.0002$ | $2.1595 \pm 0.0040$ |
| Gen-CNP | $\mathbf{0.0920 \pm 0.0002}$ | $\mathbf{0.1350 \pm 0.0002}$ | $\mathbf{2.1879 \pm 0.0048}$ |
| NP | $0.0935 \pm 0.0004$ | $0.1373 \pm 0.0005$ | $2.4811 \pm 0.0147$ |
| Gen-NP | $\mathbf{0.0933 \pm 0.0002}$ | $\mathbf{0.1369 \pm 0.0003}$ | $\mathbf{2.5237 \pm 0.0075}$ |
| ANP | $0.0763 \pm 0.0002$ | $0.1182 \pm 0.0004$ | $2.9209 \pm 0.0037$ |
| Gen-ANP | $\mathbf{0.0759 \pm 0.0001}$ | $\mathbf{0.1176 \pm 0.0001}$ | $\mathbf{2.9634 \pm 0.0077}$ |
| BNP | $0.0926 \pm 0.0004$ | $0.1340 \pm 0.0003$ | $2.7691 \pm 0.0025$ |
| Gen-BNP | $\mathbf{0.0900 \pm 0.0002}$ | $\mathbf{0.1314 \pm 0.0002}$ | $\mathbf{2.7758 \pm 0.0030}$ |
| TNP | $0.0754 \pm 0.0002$ | $0.1146 \pm 0.0002$ | $4.4044 \pm 0.0201$ |
| Gen-TNP | $\mathbf{0.0753 \pm 0.0001}$ | $\mathbf{0.1144 \pm 0.0000}$ | $\mathbf{4.4086 \pm 0.0081}$ |

Table 6: Comparison of Gen-NPs vs the baselines on EMNIST dataset with various evaluation metrics. We train 5 instances with different seeds for each method and report the mean and std. We evaluate on both seen and unseen classes.

| Setting | Method | MAE | RMSE | Log-Likelihood |
|---|---|---|---|---|
| Seen classes (0-9) | CNP | $0.0933 \pm 0.0004$ | $0.1829 \pm 0.0006$ | $0.7373 \pm 0.0037$ |
| | Gen-CNP | $\mathbf{0.0853 \pm 0.0009}$ | $\mathbf{0.1704 \pm 0.0014}$ | $\mathbf{0.7864 \pm 0.0054}$ |
| | NP | $0.0948 \pm 0.0006$ | $0.1850 \pm 0.0007$ | $0.7954 \pm 0.0022$ |
| | Gen-NP | $\mathbf{0.0900 \pm 0.0009}$ | $\mathbf{0.1793 \pm 0.0010}$ | $\mathbf{0.8142 \pm 0.0063}$ |
| | ANP | $0.0681 \pm 0.0008$ | $0.1425 \pm 0.0011$ | $0.9808 \pm 0.0060$ |
| | Gen-ANP | $\mathbf{0.0673 \pm 0.0006}$ | $\mathbf{0.1411 \pm 0.0008}$ | $\mathbf{0.9865 \pm 0.0043}$ |
| | BNP | $0.0926 \pm 0.0009$ | $0.1803 \pm 00013$ | $0.8699 \pm 0.0054$ |
| | Gen-BNP | $\mathbf{0.0828 \pm 0.0014}$ | $\mathbf{0.1653 \pm 0.0020}$ | $\mathbf{0.9051 \pm 0.0063}$ |
| | TNP | $0.0585 \pm 0.0008$ | $0.1231 \pm 0.0008$ | $1.5502 \pm 0.0036$ |
| | Gen-TNP | $\mathbf{0.0578 \pm 0.0004}$ | $\mathbf{0.1221 \pm 0.0009}$ | $\mathbf{1.5550 \pm 0.0021}$ |
| Unseen classes (10-46) | CNP | $0.1231 \pm 0.0005$ | $0.2264 \pm 0.0007$ | $0.4854 \pm 0.0035$ |
| | Gen-CNP | $\mathbf{0.1098 \pm 0.0013}$ | $\mathbf{0.2084 \pm 0.0013}$ | $\mathbf{0.5556 \pm 0.0055}$ |
| | NP | $0.1261 \pm 0.0014$ | $0.2306 \pm 0.0013$ | $0.5840 \pm 00033$ |
| | Gen-NP | $\mathbf{0.1184 \pm 0.0009}$ | $\mathbf{0.2218 \pm 0.0011}$ | $\mathbf{0.6031 \pm 0.0079}$ |
| | ANP | $0.0829 \pm 0.0004$ | $0.1676 \pm 0.0007$ | $0.8838 \pm 0.0030$ |
| | Gen-ANP | $\mathbf{0.0824 \pm 0.0006}$ | $\mathbf{0.1668 \pm 0.0010}$ | $\mathbf{0.8862 \pm 0.0036}$ |
| | BNP | $0.1229 \pm 0.0014$ | $0.2225 \pm 0.0020$ | $0.7156 \pm 0.0117$ |
| | Gen-BNP | $\mathbf{0.1077 \pm 0.0019}$ | $\mathbf{0.2030 \pm 0.0022}$ | $\mathbf{0.7642 \pm 0.0081}$ |
| | TNP | $0.0707 \pm 0.0013$ | $0.1452 \pm 0.0013$ | $1.4190 \pm 0.0061$ |
| | Gen-TNP | $\mathbf{0.0701 \pm 0.0001}$ | $\mathbf{0.1447 \pm 0.0002}$ | $\mathbf{1.4232 \pm 0.0045}$ |

## C.4 Bayesian Optimization

The following sections provide a detailed description of the training and evaluation processes for the Bayesian optimization task.

**Training**   In the training phase, the 1D scenario follows the approach outlined in Section 4.1. For multi-dimensional input $x$, training data is generated using the method proposed by NPs, where multivariate Gaussian Processes (GPs) with a Radial Basis Function (RBF) kernel are employed. In the 2D scenario, the number of total points $N$ is sampled from a uniform distribution $\mathcal{U}[60, 128]$, and the number of context points $m$ is sampled from $\mathcal{U}[30, 98]$. Similarly, in the 3D scenario, $N$ is sampled from $\mathcal{U}[128, 256]$, and $m$ is sampled from $\mathcal{U}[64, 192]$. This training setup ensures that the models are

well-prepared to handle the complexities of Bayesian optimization tasks in both one-dimensional and multi-dimensional spaces.

**Evaluation**   For the evaluation phase, in the 1D scenario, the objective functions are generated from Gaussian Processes with RBF, Matérn 5/2, and Periodic kernels. In multi-dimensional settings, various benchmark functions from the optimization literature are employed, with the Bayesian optimization process implemented using a comprehensive framework that includes both the optimization and acquisition functions. Each objective function undergoes 100 iterations of Bayesian optimization, with simple regret serving as the primary evaluation metric. This metric provides a clear indication of the model's performance by measuring the difference between the best-known value and the actual value found during optimization.

**Results:**   In the main body of the paper, for the sake of clarity, we only presented the results for Bayesian optimization using CNP, NP, and ANP methods. However, to provide a more comprehensive and detailed comparison, Figures 9 to 11 showcase the results for all methods, including their Gen-enhanced variants, across different dimensions and benchmark functions. Each method's performance is individually illustrated for 1D, 2D, and 3D Bayesian optimization tasks, providing a clearer visualization of the differences.

Figure 9 focuses on the 1D Bayesian optimization tasks, where results for different kernel functions (RBF, Matérn 5/2, and Periodic) are presented for each method. Figure 10 expands the comparison to 2D tasks, demonstrating the performance across various benchmark functions like Ackley, Dropwave, and Michalewicz. Finally, Figure 11 extends the analysis to 3D tasks, further illustrating the effectiveness of each method on more complex functions such as Cosine and Rastrigin.

These figures collectively offer a detailed and nuanced understanding of how each method performs under varying conditions, highlighting the consistent advantages of the Gen-enhanced approaches in achieving lower regret and more stable optimization results across all tested scenarios.

Table 7: Cumulative regret for different methods across various $\delta$ values.

| Method | $\delta = 0.7$ | $\delta = 0.9$ | $\delta = 0.95$ | $\delta = 0.99$ | $\delta = 0.995$ | $\delta = 0.999$ |
|---|---|---|---|---|---|---|
| Uniform | $100.00 \pm 1.18$ | $100.00 \pm 3.03$ | $100.00 \pm 4.16$ | $100.00 \pm 7.52$ | $100.00 \pm 8.11$ | $100.00 \pm 7.96$ |
| CNP | $1.66 \pm 0.14$ | $8.86 \pm 0.56$ | $8.31 \pm 0.85$ | $23.84 \pm 0.58$ | $34.10 \pm 0.56$ | $83.90 \pm 1.97$ |
| Gen-CNP | $\mathbf{1.38 \pm 0.19}$ | $\mathbf{4.57 \pm 0.55}$ | $\mathbf{6.41 \pm 0.60}$ | $\mathbf{15.85 \pm 0.73}$ | $\mathbf{22.12 \pm 0.63}$ | $\mathbf{54.46 \pm 1.50}$ |
| NP | $\mathbf{1.53 \pm 0.20}$ | $4.24 \pm 0.53$ | $5.26 \pm 0.31$ | $20.34 \pm 0.73$ | $28.85 \pm 0.43$ | $71.09 \pm 1.01$ |
| Gen-NP | $1.57 \pm 0.20$ | $\mathbf{3.11 \pm 0.22}$ | $\mathbf{4.35 \pm 0.28}$ | $\mathbf{18.63 \pm 0.93}$ | $\mathbf{26.14 \pm 0.56}$ | $\mathbf{63.96 \pm 1.44}$ |
| ANP | $103.11 \pm 44.89$ | $122.55 \pm 3.41$ | $119.75 \pm 0.87$ | $100.04 \pm 1.00$ | $89.80 \pm 1.34$ | $51.59 \pm 16.58$ |
| Gen-ANP | $\mathbf{87.77 \pm 54.56}$ | $\mathbf{96.40 \pm 49.09}$ | $\mathbf{98.22 \pm 40.54}$ | $\mathbf{91.78 \pm 11.25}$ | $\mathbf{84.40 \pm 7.71}$ | $\mathbf{44.98 \pm 12.70}$ |
| BNP | $74.04 \pm 1.75$ | $77.46 \pm 2.05$ | $74.55 \pm 2.56$ | $87.88 \pm 4.43$ | $97.62 \pm 5.15$ | $108.79 \pm 4.86$ |
| Gen-BNP | $\mathbf{42.68 \pm 1.35}$ | $\mathbf{34.42 \pm 2.28}$ | $\mathbf{24.00 \pm 2.45}$ | $\mathbf{27.65 \pm 4.51}$ | $\mathbf{34.37 \pm 4.64}$ | $\mathbf{59.00 \pm 3.58}$ |
| TNP | $3.02 \pm 2.52$ | $3.27 \pm 1.24$ | $5.76 \pm 2.03$ | $19.61 \pm 2.84$ | $27.67 \pm 3.83$ | $9.61 \pm 2.80$ |
| Gen-TNP | $\mathbf{1.91 \pm 1.09}$ | $\mathbf{1.85 \pm 0.43}$ | $\mathbf{2.63 \pm 0.50}$ | $\mathbf{3.18 \pm 0.89}$ | $\mathbf{4.58 \pm 1.85}$ | $\mathbf{8.89 \pm 0.39}$ |

## C.5   Contextual bandits

The study compares Gen-NPs with baselines using the wheel bandit framework. This framework involves a unit circle segmented into a low-reward zone (colored blue) and four high-reward zones of different colors. The division is controlled by a scalar $\delta$, which sets the boundary of the low-reward zone, leaving the other four zones equally sized. An agent, unaware of $\delta$'s value, selects from five potential actions based on its position within the circle.

When the agent's position $\|X\|$ is less than or equal to $\delta$, it is located in the low-reward zone. The optimal choice here is action $k = 1$, rewarding the agent with $r \sim \mathcal{N}(1.2, 0.01^2)$. All other actions yield $r \sim \mathcal{N}(1.0, 0.01^2)$. Conversely, if $\|X\| > \delta$, indicating presence in a high-reward zone, the agent should choose from actions $k = 2 - 5$. These choices can grant a significant reward of $r \sim \mathcal{N}(50.0, 0.01^2)$, with all non-optimal choices returning $\mathcal{N}(1.0, 0.01^2)$, except for $k = 1$.

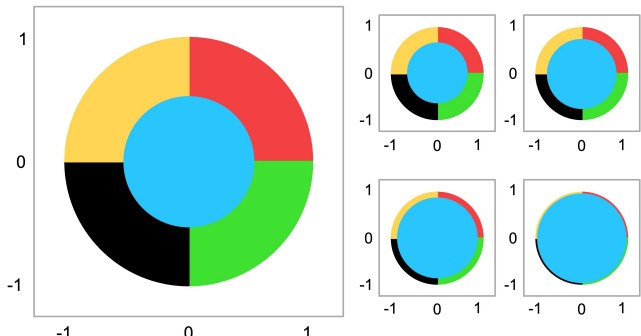

Figure 12: The wheel bandit problem with varying values of $\delta$.

**Training** A dataset is created by generating $B$ different wheel problem instances $\{\delta_i\}_{i=1}^B$, where $\delta$ values are uniformly distributed between 0 and 1. For each instance, $N$ points are sampled to assess and select $m$ points as context for training, with each point being a pair $(X, r)$ of coordinates and the corresponding reward. The goal is to learn to predict reward values based on $X$. Parameters are set with $B = 8$, $N = 562$, and $m = 512$.

**Evaluation** Experiments are conducted by testing the Gen-NPs and baseline approaches across varying $\delta$ values, using 50 different seeds for each setting. Over 2000 steps per trial, each agent's task is to estimate the reward values for five strategies based on $X$, choose according to the Upper Confidence Bound (UCB) strategy, and receive the actual reward for the selected strategy. Cumulative regret is utilized to measure the performance effectiveness.

**Results** As shown in Table 7, for cumulative regret, the Gen versions of CNP, NP, ANP, BNP, and TNP generally achieve lower regret across all $\delta$ settings, demonstrating their superior capability in handling the complex decision-making scenarios introduced by the wheel bandit framework. Notably, Gen-TNP outperforms all other methods, especially in the most difficult cases (higher $\delta$), where it maintains low regret with minimal variance.

Similarly, Table 8 presents the simple regret outcomes, further confirming the advantage of Gen-enhanced methods. The simple regret is notably lower for Gen versions, indicating more effective exploration and exploitation of the reward landscape, which is crucial in achieving optimal decision-making.

The visual depiction of the wheel bandit problem with varying $\delta$ values is provided in Figure 12. This figure illustrates the segmentation of the reward zones within the unit circle, helping to contextualize the challenge faced by the models in predicting optimal actions based on incomplete and uncertain information.

Overall, the results clearly demonstrate the effectiveness of integrating the proposed general recipe into neural process-based models, significantly improving their performance in contextual bandit problems by reducing both cumulative and simple regret across various scenarios.

Table 8: Simple regret for different methods across various $\delta$ values.

| Method | $\delta = 0.7$ | $\delta = 0.9$ | $\delta = 0.95$ | $\delta = 0.99$ | $\delta = 0.995$ | $\delta = 0.999$ |
|---|---|---|---|---|---|---|
| Uniform | $100.00 \pm 20.77$ | $100.00 \pm 34.60$ | $100.00 \pm 50.34$ | $100.00 \pm 96.59$ | $100.00 \pm 114.30$ | $100.00 \pm 120.11$ |
| CNP | $1.43 \pm 2.24$ | $9.27 \pm 10.13$ | $8.59 \pm 9.85$ | $24.70 \pm 1.36$ | $34.82 \pm 1.88$ | $83.05 \pm 4.13$ |
| Gen-CNP | $\mathbf{1.08 \pm 1.75}$ | $\mathbf{4.50 \pm 6.37}$ | $\mathbf{6.07 \pm 8.36}$ | $\mathbf{16.14 \pm 2.04}$ | $\mathbf{22.38 \pm 2.76}$ | $\mathbf{53.43 \pm 6.48}$ |
| NP | $1.42 \pm 2.14$ | $3.78 \pm 5.28$ | $4.95 \pm 5.59$ | $20.85 \pm 1.81$ | $29.40 \pm 2.57$ | $70.34 \pm 5.64$ |
| Gen-NP | $\mathbf{1.30 \pm 2.06}$ | $\mathbf{2.89 \pm 4.18}$ | $\mathbf{4.20 \pm 3.52}$ | $\mathbf{19.10 \pm 1.94}$ | $\mathbf{26.82 \pm 2.68}$ | $\mathbf{63.61 \pm 6.17}$ |
| ANP | $104.60 \pm 15.58$ | $125.37 \pm 36.50$ | $122.68 \pm 55.36$ | $104.02 \pm 112.00$ | $95.26 \pm 134.43$ | $44.79 \pm 144.48$ |
| Gen-ANP | $\mathbf{88.46 \pm 14.62}$ | $\mathbf{98.99 \pm 32.93}$ | $\mathbf{101.83 \pm 51.37}$ | $\mathbf{101.29 \pm 111.32}$ | $\mathbf{93.91 \pm 132.74}$ | $\mathbf{40.21 \pm 129.59}$ |
| BNP | $70.24 \pm 15.58$ | $73.72 \pm 28.81$ | $71.70 \pm 40.01$ | $84.53 \pm 84.77$ | $95.24 \pm 101.92$ | $106.83 \pm 100.60$ |
| Gen-BNP | $\mathbf{42.81 \pm 12.88}$ | $\mathbf{33.15 \pm 19.68}$ | $\mathbf{22.24 \pm 21.15}$ | $\mathbf{23.24 \pm 31.65}$ | $\mathbf{31.23 \pm 41.65}$ | $\mathbf{60.13 \pm 65.35}$ |
| TNP | $1.37 \pm 2.11$ | $2.29 \pm 4.32$ | $4.70 \pm 10.05$ | $17.75 \pm 41.28$ | $25.52 \pm 59.07$ | $8.63 \pm 3.67$ |
| Gen-TNP | $\mathbf{1.05 \pm 1.57}$ | $\mathbf{1.42 \pm 2.86}$ | $\mathbf{2.24 \pm 5.46}$ | $\mathbf{2.96 \pm 1.22}$ | $\mathbf{4.00 \pm 1.65}$ | $\mathbf{8.68 \pm 3.84}$ |

## C.6 Ablation Study

We conducted ablation studies to validate the effectiveness of the proposed *Risk-Aware Dynamical Stability Regularization (DSR)* and *Optimization-Aware Noise Injection Learning Strategy (NILS)* on 1D regression task. Table 9 presents the results of the original method, Gen-NPs with only DSR, Gen-NPs with only NILS, and the full Gen-NPs with both modules included.

Table 9: Ablation study results comparing the original method.

| Method | LL (RBF) | GI (RBF) | LL (Periodic) | GI (Periodic) |
|---|---|---|---|---|
| Original CNP | $0.265_{\pm 0.015}$ | $0.880_{\pm 0.027}$ | $-1.435_{\pm 0.020}$ | $1.312_{\pm 0.053}$ |
| Gen-CNP (with DSR only) | $0.276_{\pm 0.013}$ | $0.858_{\pm 0.030}$ | $-1.428_{\pm 0.022}$ | $1.202_{\pm 0.045}$ |
| Gen-CNP (with NILS only) | $0.279_{\pm 0.012}$ | $0.846_{\pm 0.035}$ | $-1.423_{\pm 0.024}$ | $1.151_{\pm 0.041}$ |
| Full Gen-CNP (DSR + NILS) | $\mathbf{0.286}_{\pm \mathbf{0.010}}$ | $\mathbf{0.830}_{\pm \mathbf{0.042}}$ | $\mathbf{-1.418}_{\pm \mathbf{0.023}}$ | $\mathbf{1.112}_{\pm \mathbf{0.039}}$ |
| Original NP | $0.240_{\pm 0.022}$ | $0.490_{\pm 0.025}$ | $-1.145_{\pm 0.032}$ | $1.192_{\pm 0.043}$ |
| Gen-NP (with DSR only) | $0.261_{\pm 0.018}$ | $0.479_{\pm 0.022}$ | $-1.135_{\pm 0.030}$ | $1.179_{\pm 0.040}$ |
| Gen-NP (with NILS only) | $0.259_{\pm 0.014}$ | $0.480_{\pm 0.017}$ | $-1.133_{\pm 0.027}$ | $1.182_{\pm 0.038}$ |
| Full Gen-NP (DSR + NILS) | $\mathbf{0.270}_{\pm \mathbf{0.009}}$ | $\mathbf{0.470}_{\pm \mathbf{0.012}}$ | $\mathbf{-1.125}_{\pm \mathbf{0.024}}$ | $\mathbf{1.165}_{\pm \mathbf{0.036}}$ |

As shown in Table 9 and Appendix, these results highlight the significance of incorporating DSR for improving dynamical stability and NILS for robust optimization. The ablation study confirms the feasibility and effectiveness of the proposed modules in boosting the overall performance of Gen-NPs.

## C.7 Comparison with Stability Neural Processes

To provide a comprehensive analysis, we conducted additional experiments to compare stability-based generalization error (SGE) with our proposed Gen-NPs, emphasizing the differences in their noise introduction mechanisms and evaluation methods.

While SGE quantifies generalization as the difference between test error and training error, our approach focuses on gradient incoherence (GI) and directly modeling the parameter-data mutual information through controlled noise injection. To test robustness, we added Gaussian noise (mean = 0, variance = 1) to a random selection of 5% and 10% of the training data. The experiments were performed on a one-dimensional regression task with data generated using an RBF kernel.

Table 10 presents detailed results for CNP, ANP, and BNP models under original, 5% noise, and 10% noise conditions. Gen-NPs consistently achieves higher log-likelihood (LL) values and lower gradient incoherence (GI) compared to baseline methods across all noise levels. Notably, even as noise increases, Gen-NPs maintains better performance relative to standard NPs, demonstrating its enhanced robustness.

The key difference lies in how noise is utilized: Stability Neural Processes approaches add noise to the training data to measure stability, while Gen-NPs introduces noise strategically during the parameter update process. This fundamental difference enables Gen-NPs to better capture the mutual information between model parameters and training data, leading to improved generalization capacity under various conditions.

# D   Computational Complexity Analysis

In this section, we analyze the computational complexity and overhead introduced by our Gen-NPs approach compared to standard NP methods. While noise injection and dynamical stability regularization enhance model performance, they also introduce additional computation. Here we quantify these costs and demonstrate that the performance benefits outweigh the computational overhead.

**Theoretical Complexity Analysis**   Let $d$ denote the model parameter dimension, $n$ the number of context points, $m$ the number of tasks, and $b$ the batch size.

For the dynamical stability regularization (DSR) term computation, we avoid explicitly forming the full Hessian matrix, which would require $\mathcal{O}(d^3)$ operations. Instead, we utilize efficient Hessian-vector product approximations, reducing the complexity to $\mathcal{O}(bd^2)$. The noise injection component

Table 10: Comparison of log-likelihood (LL), gradient incoherence (GI), and stability-based generalization error (SGE) under different noise levels. Results are reported as mean ± standard deviation.

| Metric | Method | LL | GI | SGE |
|---|---|---|---|---|
| Original | CNP | 0.272 ± 0.013 | 0.865 ± 0.025 | 0.872 ± 0.030 |
| | **Gen-CNP** | **0.283 ± 0.011** | **0.834 ± 0.044** | **0.850 ± 0.051** |
| | ANP | 0.809 ± 0.004 | 0.967 ± 0.043 | 1.256 ± 0.046 |
| | **Gen-ANP** | **0.810 ± 0.003** | **0.954 ± 0.014** | **1.147 ± 0.018** |
| | BNP | 0.394 ± 0.015 | 0.151 ± 0.006 | 0.872 ± 0.009 |
| | **Gen-BNP** | **0.402 ± 0.010** | **0.150 ± 0.009** | **0.783 ± 0.011** |
| Noise (+5%) | CNP | 0.234 ± 0.016 | 0.860 ± 0.020 | 0.875 ± 0.028 |
| | **Gen-CNP** | **0.271 ± 0.013** | **0.828 ± 0.042** | **0.845 ± 0.049** |
| | ANP | 0.773 ± 0.007 | 0.963 ± 0.040 | 0.975 ± 0.045 |
| | **Gen-ANP** | **0.795 ± 0.006** | **0.950 ± 0.012** | **0.960 ± 0.016** |
| | BNP | 0.362 ± 0.018 | 0.149 ± 0.005 | 0.157 ± 0.008 |
| | **Gen-BNP** | **0.388 ± 0.013** | **0.136 ± 0.008** | **0.155 ± 0.010** |
| Noise (+10%) | CNP | 0.220 ± 0.017 | 0.855 ± 0.018 | 0.870 ± 0.026 |
| | **Gen-CNP** | **0.258 ± 0.016** | **0.826 ± 0.041** | **0.838 ± 0.047** |
| | ANP | 0.750 ± 0.010 | 0.960 ± 0.038 | 0.970 ± 0.043 |
| | **Gen-ANP** | **0.765 ± 0.009** | **0.945 ± 0.011** | **0.955 ± 0.014** |
| | BNP | 0.340 ± 0.021 | 0.146 ± 0.004 | 0.155 ± 0.007 |
| | **Gen-BNP** | **0.375 ± 0.015** | **0.138 ± 0.007** | **0.153 ± 0.009** |

Table 11: Theoretical complexity comparison between standard NPs and Gen-NPs

| Operation | Standard NP | Gen-NP (Ours) |
|---|---|---|
| Forward pass | $\mathcal{O}(bnd)$ | $\mathcal{O}(bnd)$ |
| Backward pass | $\mathcal{O}(bd^2)$ | $\mathcal{O}(bd^2)$ |
| DSR computation | – | $\mathcal{O}(bd^2)$ |
| Noise injection | – | $\mathcal{O}(d)$ |

has minimal overhead of $\mathcal{O}(d)$ for sampling from a Gaussian distribution and adding the noise to the parameter updates.

**Empirical Evaluation** We measured the actual computational overhead across different tasks using the same hardware setup for all experiments. Table 12 summarizes these findings.

Table 12: Empirical computational overhead and performance gains

| Method | Training Time | Memory Usage | Avg. LL Improvement |
|---|---|---|---|
| Original CNP | 1.00× | 1.00× | – |
| Gen-CNP (with DSR only) | 1.12× | 1.08× | +4.1% |
| Gen-CNP (with NILS only) | 1.05× | 1.02× | +5.3% |
| Full Gen-CNP (DSR + NILS) | 1.18× | 1.10× | +7.9% |

**Task-Specific Training Time Analysis** We further analyzed the training time across different tasks and architectures to provide a comprehensive view of the computational overhead.

**Cost-Benefit Analysis** While Gen-NPs introduce approximately 10% additional training time, the consistent performance improvements across all tasks and metrics easily justify this minimal overhead. The most computationally intensive component is the DSR term, specifically the computation of Hessian-related properties. However, this overhead is only present during training; inference time remains virtually identical to standard NP methods.

Table 13: Training time comparison across different tasks (in hours)

| Method | 1D Regression | Image Completion | Bayesian Optimization |
|---|---|---|---|
| CNP | 0.33 | 2.45 | 0.86 |
| Gen-CNP | 0.36 | 2.68 | 0.95 |
| NP | 0.41 | 2.98 | 1.04 |
| Gen-NP | 0.45 | 3.28 | 1.14 |
| ANP | 0.58 | 4.12 | 1.48 |
| Gen-ANP | 0.64 | 4.53 | 1.62 |

For tasks requiring high accuracy and reliable uncertainty quantification, such as Bayesian optimization and medical image completion, the 7-9% improvement in log-likelihood represents a significant practical advantage that substantially outweighs the modest 10% increase in training resources.

Furthermore, we found that in practice, Gen-NPs often require fewer training iterations to reach a target performance level compared to standard NPs, which can fully offset the per-iteration computational overhead in end-to-end training scenarios. This favorable performance-to-cost ratio makes Gen-NPs particularly attractive for practical applications where generalization and reliable uncertainty estimation are critical.

**Implementation Considerations**   To minimize the computational overhead while maintaining performance benefits, we recommend:

1. Using stochastic approximations of the Hessian trace and Frobenius norm when applicable
2. Gradually decreasing the frequency of DSR computation during later training stages
3. Implementing the DSR term computation with efficient auto-differentiation libraries that optimize Hessian-vector products
4. For very large models, considering a reduced-precision implementation of the DSR component

These optimizations can further reduce the computational gap between standard NPs and Gen-NPs while preserving the generalization benefits of our approach.

# E   Algorithm Pseudocode

We present the complete algorithm for Generalization Neural Processes (Gen-NPs) that integrates both the Risk-Aware Dynamical Stability Regularization (DSR) and Optimization-Aware Noise Injection Learning Strategy (NILS) components. Algorithm 1 provides a comprehensive pseudocode implementation that practitioners can follow to apply our method to various Neural Process variants.

---

**Algorithm 1** Generalization Neural Processes (Gen-NPs)

---

**Require:** Task environment $\tau$, initial learning rate $\eta_0$, inverse temperature $\gamma$, number of tasks per batch $B$, number of iterations $S$, DSR coefficients $\lambda_1 \in [0.01, 0.1]$, $\lambda_2 \in [0.001, 0.01]$

1: Randomly initialize $\theta^0$
2: **for** $s \leftarrow 1$ to $S$ **do**
3:     Sample a batch of tasks $\{\mathcal{D}_i\}_{i=1}^{B} \sim \tau$
4:     Initialize task gradients and DSR term: $G(\theta^{s-1}) = 0$, $R_{\mathrm{dyn}} = 0$
5:     **for** each task $i \in [B]$ **do**
6:         Randomly split task $\mathcal{D}_i$ into context set $\mathcal{D}_i^C$ and target set $\mathcal{D}_i^T$
7:         Calculate task-specific empirical risk $\tilde{R}_{\mathcal{D}_i^T}(\theta^{s-1})$
8:         Calculate task-specific gradient $g_i(\theta^{s-1}, \mathcal{D}_i^C, \mathcal{D}_i^T)$
9:         Estimate Hessian trace $Tr(H_i)$ and Frobenius norm $\|H_i\|_F$ using efficient approximations
10:         Update DSR term: $R_{\mathrm{dyn}} += \frac{1}{B}(\lambda_1 \cdot Tr(H_i) + \lambda_2 \cdot \|H_i\|_F)$
11:     **end for**
12:     Aggregate gradients over all tasks: $G(\theta^{s-1}) = \frac{1}{B}\sum_{i=1}^{B} g_i(\theta^{s-1}, \mathcal{D}_i^C, \mathcal{D}_i^T)$
13:     Calculate gradient of DSR term: $\nabla R_{\mathrm{dyn}}$
14:     Update learning rate to $\eta_s$
15:     Calculate Gaussian noise variance $\sigma_s^2 = \frac{\eta_s}{\gamma}$
16:     Sample Gaussian noise $\xi^s \sim \mathcal{N}(0, \sigma_s^2 I_k)$
17:     Update parameter $\theta^s = \theta^{s-1} - \eta_s(G(\theta^{s-1}) + \nabla R_{\mathrm{dyn}}) + \xi^s$
18: **end for**

---

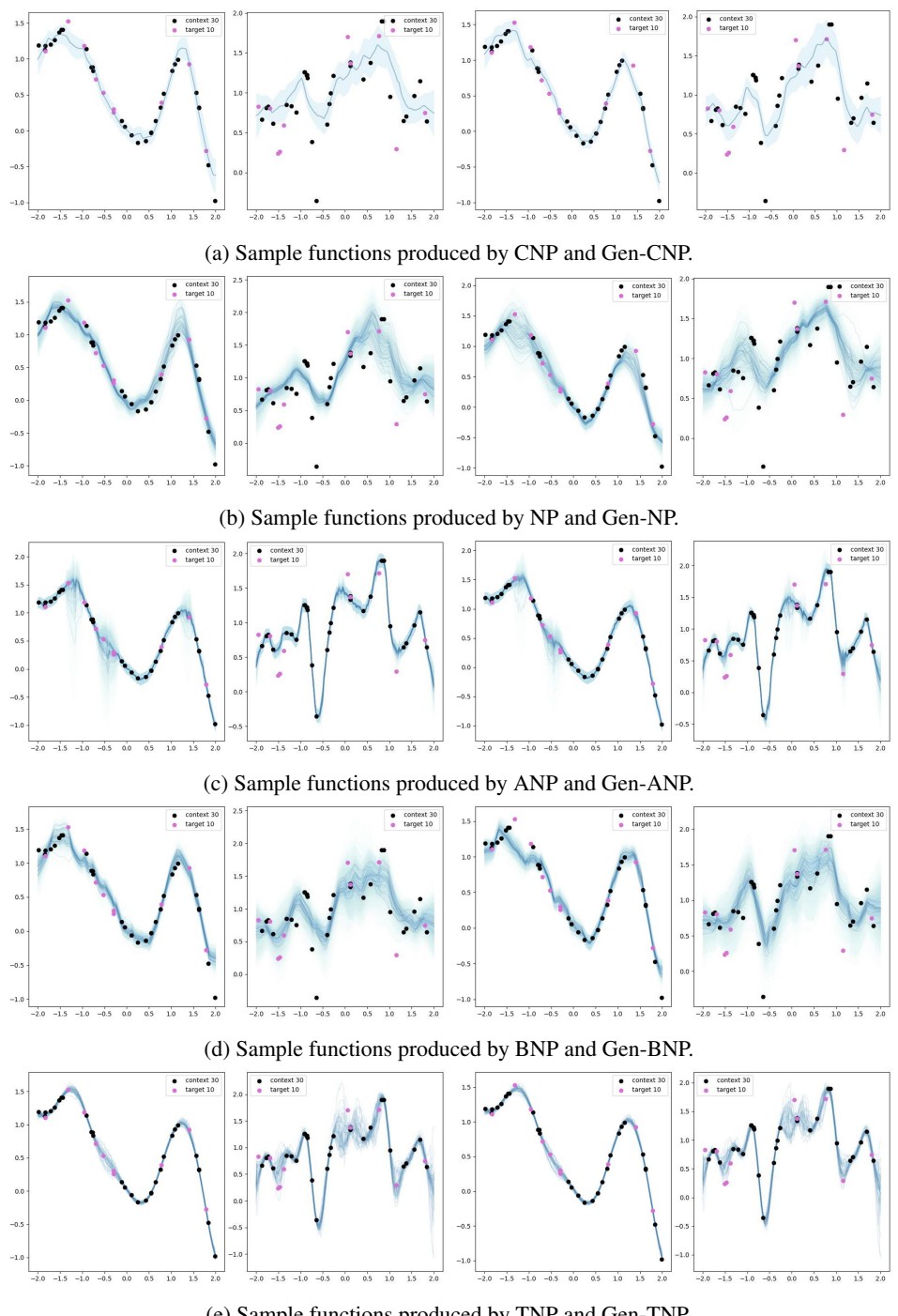

(a) Sample functions produced by CNP and Gen-CNP.

(b) Sample functions produced by NP and Gen-NP.

(c) Sample functions produced by ANP and Gen-ANP.

(d) Sample functions produced by BNP and Gen-BNP.

(e) Sample functions produced by TNP and Gen-TNP.

Figure 6: Sample functions produced by NPs and their corresponding Gen variants given 30 context points. Data is generated from a GP with an RBF kernel. Each solid blue curve corresponds to one sample function, and the blue area around each curve represents the variance in the predictive distribution. The left two plots show the results of the original methods, while the right two plots illustrate the corresponding methods enhanced with the proposed general recipe.

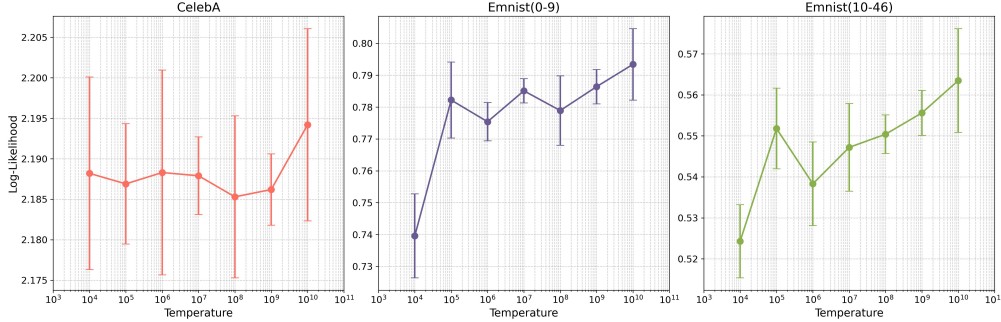

Figure 7: Experimental results across different temperature values. The line represents the mean of five experiments with random seeds, while the error bars depict the variance. The three subplots correspond to the log-likelihood results for CelebA and EMNIST datasets.

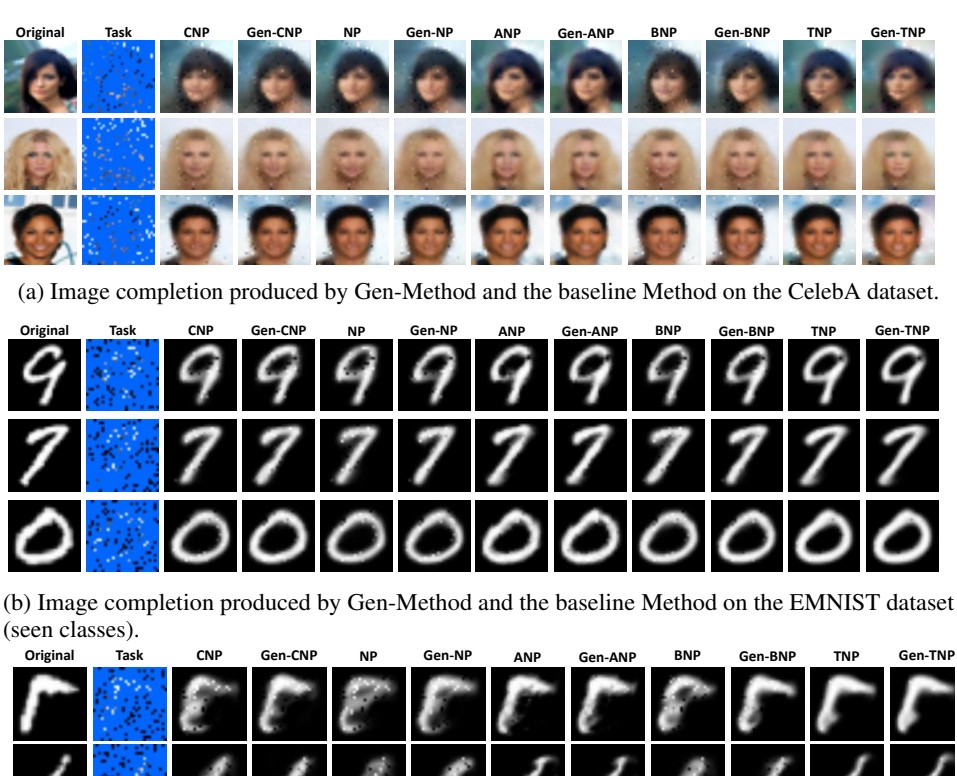

(a) Image completion produced by Gen-Method and the baseline Method on the CelebA dataset.

(b) Image completion produced by Gen-Method and the baseline Method on the EMNIST dataset (seen classes).

(c) Image completions produced by Gen-Method and the baseline Method on the EMNIST dataset (unseen classes).

Figure 8: Image completion produced by Gen-Method and the baseline Method methods given 100 context points.

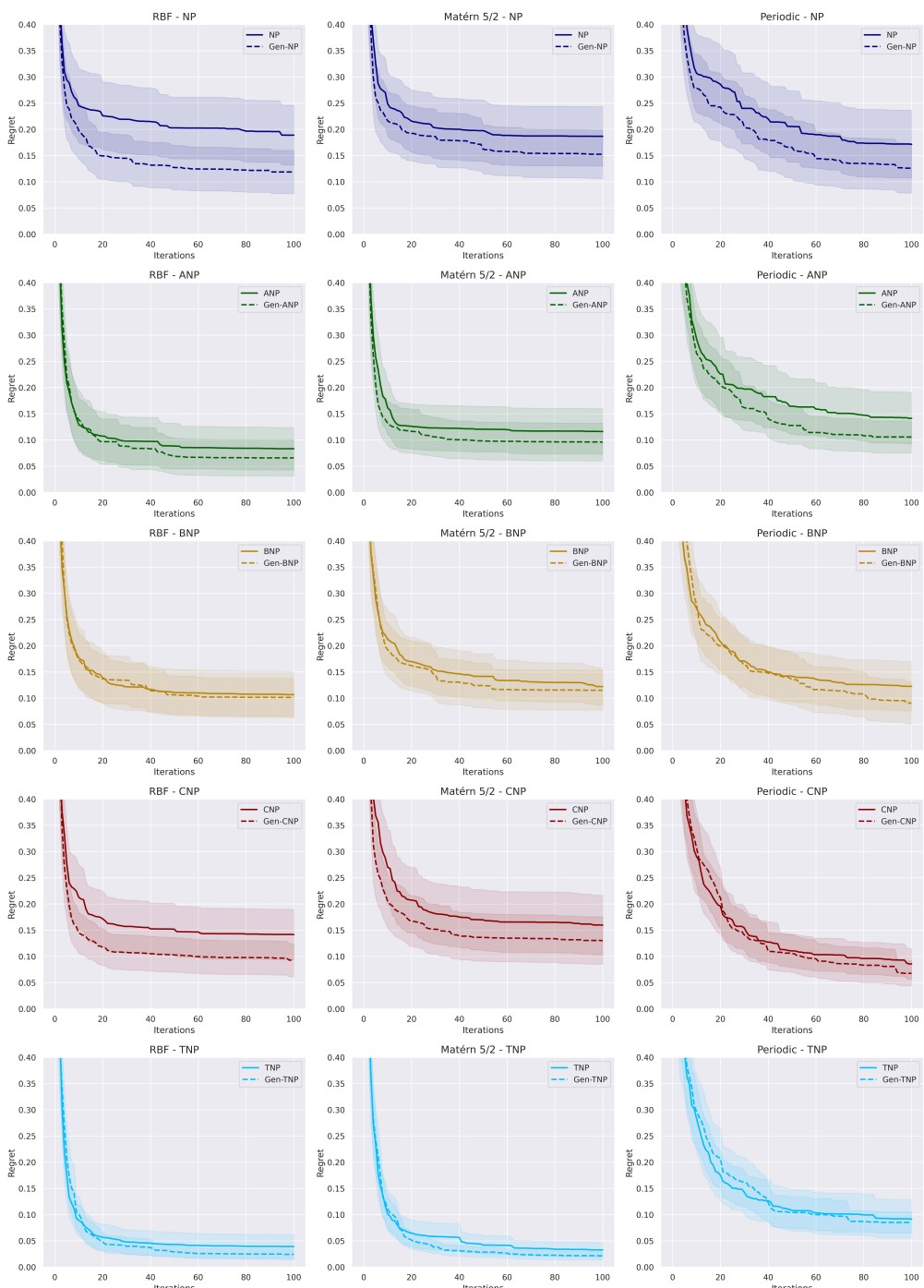

Figure 9: Regret performance on 1D Bayesian Optimization (BO) tasks.

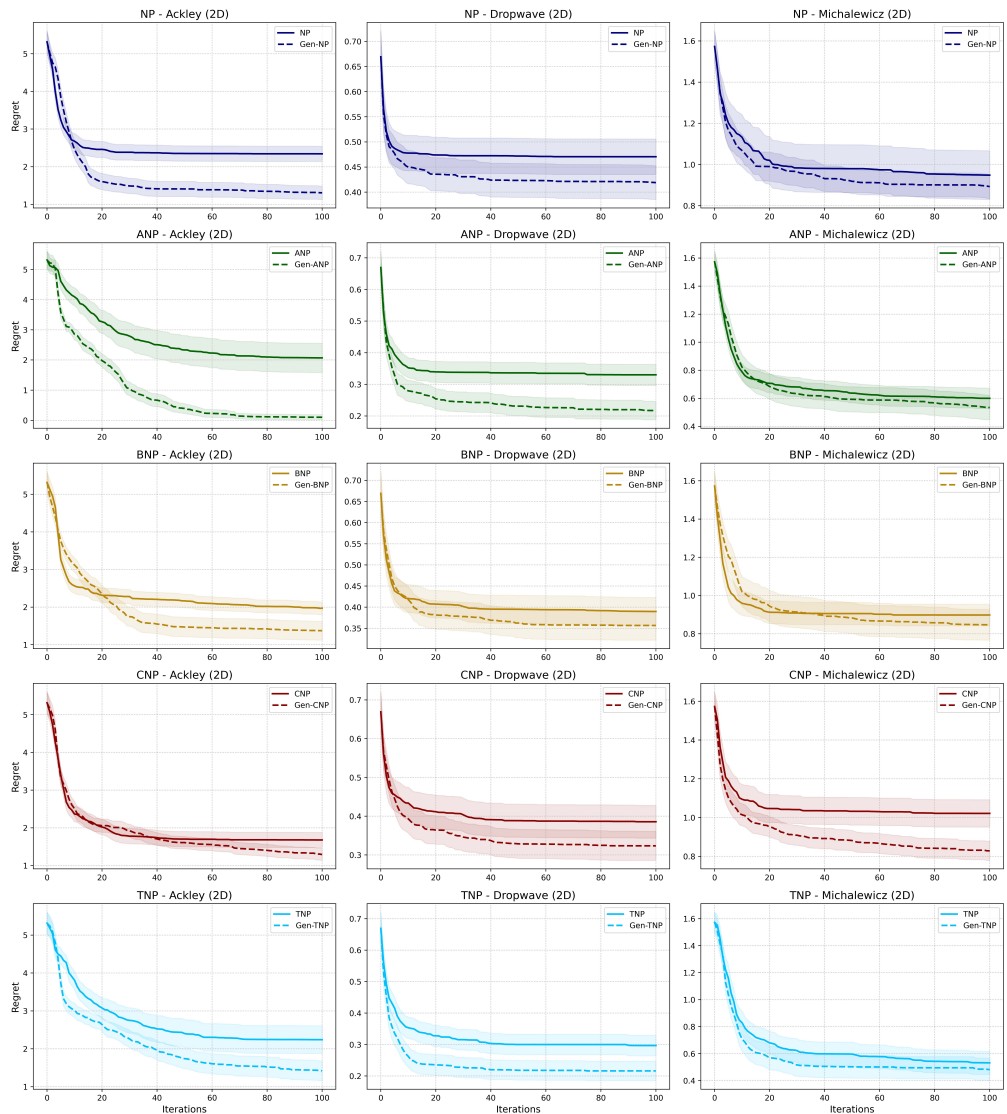

Figure 10: Regret performance on 2D Bayesian Optimization (BO) tasks.

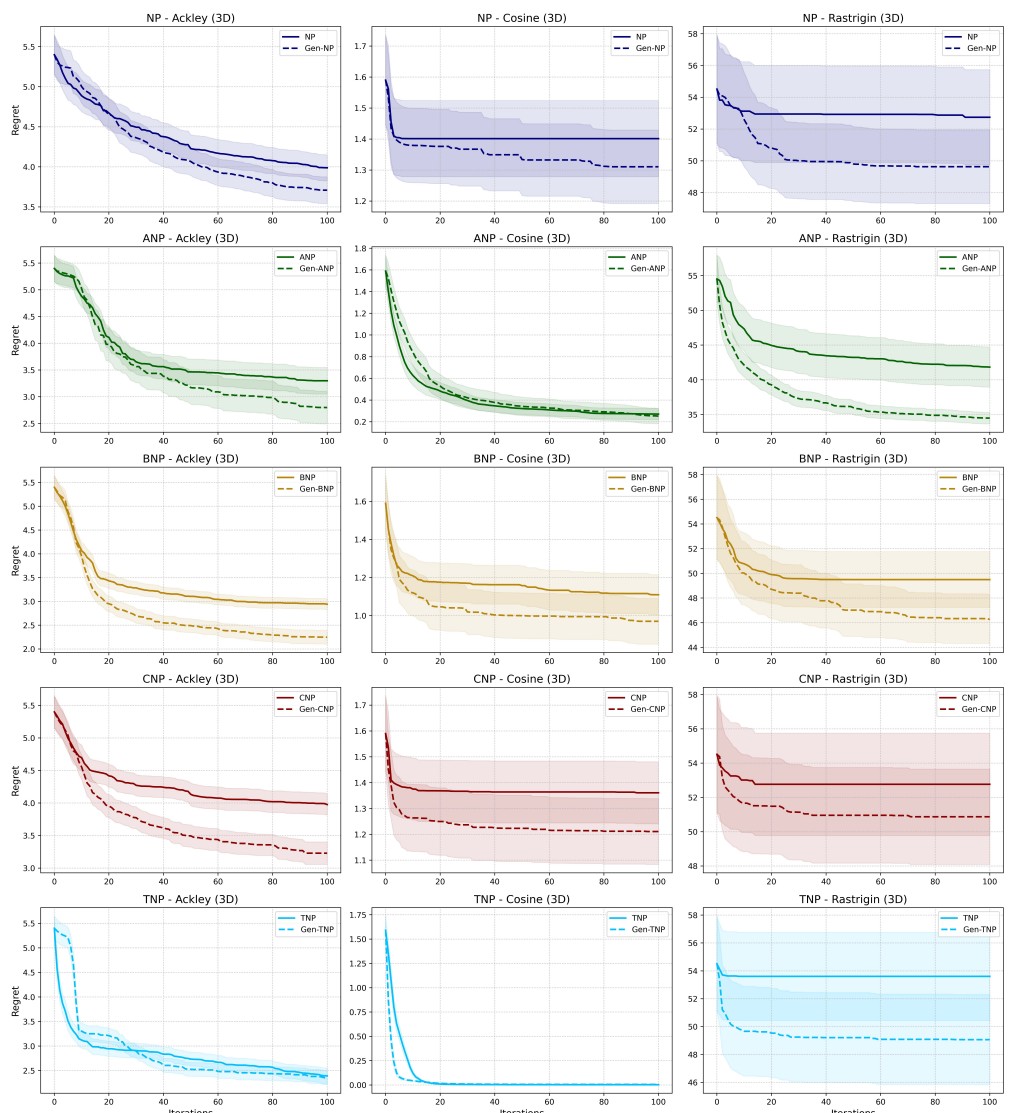

Figure 11: Regret performance on 3D Bayesian Optimization (BO) tasks.

