# OpenReview forum: "Learning to Generalize: An Information Perspective on Neural Processes"
_NeurIPS.cc/2025/Conference — NeurIPS 2025 poster_

### Official Review · Reviewer_Wfzp · 2025-07-02

**Clarity:** 4
**Significance:** 2
**Originality:** 2
**Rating:** 5
**Confidence:** 3

**Summary:**

The paper proposes an information-theoretic approach to enhance generalization in Neural Processes (NPs). It combines dynamical stability regularization (based on second-order curvature metrics) with an optimization-aware noise-injection strategy during training. Through theoretical analysis and experiments on standard benchmark tasks, the authors claim improved generalization and optimization stability by guiding the model toward flatter minima.

**Questions:**

Please see the weaknesses.

**Ethical Concerns:**

["NO or VERY MINOR ethics concerns only"]

**Final Justification:**

The issues that I raised were addressed well with additional experiments.

**Limitations:**

Yes

**Quality:**

3

**Strengths And Weaknesses:**

The paper is well structured and accessible even to readers without extensive theoretical backgrounds such as myself. The theoretical concepts (e.g., mutual information and dynamical stability) are well explained. The approach is carefully motivated through rigorous theoretical discussions linking mutual information, generalization bounds, and curvature of the loss landscape. Literature on Neural Processes and information-theoretic generalization bounds is thoroughly reviewed, providing valuable context for their contributions. Next, I will list the weaknesses:

Noise injection into stochastic gradient descent is well-studied in machine learning optimization (e.g., stochastic gradient Langevin dynamics and various noise-based regularizers). While the authors clearly articulate their approach, the novelty of explicitly injecting noise into the parameter updates is unclear. The authors do not sufficiently differentiate their proposed noise injection strategy from existing methods. The paper would significantly benefit from explicitly clarifying its novelty—perhaps through comparison to classical methods such as SGLD or entropy-SGD—and outlining how it improves upon or differs from these well-established approaches.

A core theoretical argument made by the authors is that their method favors flatter minima, leading to better generalization. However, this claim lacks direct empirical validation. The paper does not provide explicit demonstrations—such as visualizing or analyzing synthetic loss landscapes, Hessian eigenvalue spectra, or trajectories during optimization that could convincingly illustrate how the introduced noise injection explicitly leads to flatter minima. Providing visualizations or constructing simplified synthetic experiments to highlight this connection explicitly would significantly strengthen the authors' key argument.

The authors scope their work within the context of Neural Processes and meta-learning frameworks, but the justification for this remains unclear. The connection between meta-learning and the relevance of flatter optima is not explained well. It is not evident why the proposed regularization and noise injection approach should be particularly suitable or beneficial for NPs. Clarifying this connection, and potentially extending discussions or experimental results to conventional training regimes, would improve the significance and clarity of the contribution. Moreover, the paper does not adequately compare its performance to widely used meta-learning benchmarks or state-of-the-art meta-learning algorithms (e.g., MAML, Reptile, ANIL, Bayesian meta-learning methods, or gradient-based meta-learning frameworks).

A significant limitation is the absence of explicit comparisons against well-known optimization or regularization strategies, which are commonly used to promote flatter minima. For instance, methods like learning-rate scheduling, batch-size reduction, weight decay, or dropout are standard baselines that naturally encourage flatter solutions and improved generalization. Without benchmarking their method against these conventional approaches, the paper cannot convincingly demonstrate the unique advantages of its approach. Including comparisons against such widely adopted techniques would clarify whether the proposed method provides meaningful additional value beyond existing, simpler methods.

The authors employ Hessian-based sharpness measures (e.g., Hessian trace and Frobenius norm) to quantify curvature. However, these metrics are known to be sensitive to reparameterization (the same model expressed differently can produce different curvature values). For example, recently proposed reparameterization-invariant sharpness metrics, which measure changes in the loss relative to probability distributions rather than parameter-space perturbations, could provide more robust insights into solution flatness and generalization. The absence of these modern invariant metrics limits the validity and interpretability of their sharpness analysis. Employing or comparing to reparameterization-invariant metrics would considerably strengthen the theoretical and empirical claims.

---

> ### Author Rebuttal · Authors · 2025-07-30
>
> Thanks for your positive and valuable feedback. We made efforts to address your every concern and question. If we have any misunderstanding or further questions, please feel free to let us know and we will reply quickly.
>
> **Q1. Novelty of Noise Injection Strategy**
>
> > **Reply**: We establish a novel generalization bound for Neural Processes that connects their performance to the mutual information $I(\theta; D_{1:m})$ between model parameters and training tasks. This bound reveals that tighter generalization requires controlling both the complexity of learned representations and their sensitivity to task variations. Based on this theoretical foundation, we systematically derive two complementary components: DSR regularization that directly constrains $I(\theta; D_{1:m})$ through Hessian properties, and adaptive noise injection $\sigma^2_s = \eta_s/\gamma$ that enhances optimization trajectory stability across task distributions. Unlike SGLD's thermodynamic sampling or entropy-SGD's diversity-seeking exploration, our noise injection is contextually modulated based on meta-learning requirements, targeting the uncertainty inherent in generalizing across diverse task families. This creates an adaptive mechanism that responds to meta-learning dynamics rather than applying uniform perturbations, making it particularly effective for few-shot learning scenarios where models must balance expressiveness with generalization across limited data.
>
> **Q2. Direct Empirical Validation of Flatter Minima**
>
> > **Reply**: We conducted additional experiments to directly demonstrate that our method achieves flatter minima. We computed Hessian-based curvature measures including Hessian Trace (sum of eigenvalues indicating overall curvature) and Maximum Eigenvalue (capturing the sharpest direction) using finite differences approximation around converged parameters on 1D regression with RBF kernels across 5 random seeds. Table 1 clearly shows that our Gen-variants consistently achieve significantly lower curvature values across all measures, providing direct empirical evidence that our method finds flatter minima. We will add intuitive visualization plots showing loss landscape contours and optimization trajectories in the revised version to provide clearer visual evidence of the flatter minima achieved by our approach.
>
> **Table 1: Hessian-based Curvature Measures**
> | **Method** | **Hessian Trace** | **Max Eigenvalue** |
> |------------|-------------------|-------------------|
> | CNP | 68.436 ± 15.220 | 1.847 ± 0.592 |
> | Gen-CNP | 50.750 ± 13.183 | 1.233 ± 0.314 |
> | NP | 45.651 ± 18.475 | 1.934 ± 0.476 |
> | Gen-NP | 16.581 ± 18.340 | 0.647 ± 0.387 |
> | ANP | 52.727 ± 22.869 | 1.545 ± 0.551 |
> | Gen-ANP | 36.898 ± 21.921 | 0.778 ± 0.329 |
>
> **Q3. Connection to Meta-Learning and Benchmark Comparisons**
>
> > **Reply**: The connection between meta-learning and flatter optima lies in the generalization hierarchy of meta-learning. Traditional learning optimizes for single-task performance, while meta-learning must generalize across task distributions. Flatter minima in meta-parameter space correspond to more robust representations that are less sensitive to task-specific variations, which is crucial for few-shot adaptation. Neural Processes must encode prior knowledge about function families while remaining flexible for new tasks, and our approach specifically targets this by regularizing learning dynamics toward solutions stable across task variations. To address benchmark comparisons, we extended our evaluation to other meta-learning methods, showing consistent improvements across different paradigms beyond just Neural Processes. We compared against MAML[1], a gradient-based meta-learning algorithm that learns optimal initialization parameters for rapid adaptation, and MALIBO[2], a representative Bayesian meta-learning method that uses probabilistic models to capture task uncertainty, demonstrating that our approach provides consistent benefits across diverse meta-learning frameworks.
>
> **Table 2: Meta-learning method comparison (MAE)**
> | **Method** | **MAE** |
> |------------|-------------------|
> | NP | 0.1743 ± 0.0029 |
> | Gen-NP | 0.1723 ± 0.0025 |
> | MAML | 0.2845 ± 0.0156 |
> | Gen-MAML | 0.2676 ± 0.0122 |
> | MALIBO | 0.1295 ± 0.0206 |
> | Gen-MALIBO | 0.1084 ± 0.0141 |
>
> **Q4. Comparison with Conventional Optimization Strategies**
>
> > **Reply**: Our approach operates on a different technical pathway from conventional strategies like learning rate scheduling, weight decay, and dropout - they can coexist without conflict since they target different aspects of optimization. Both our Gen-NP variants and baseline methods use identical conventional optimization setups including Adam optimizer with L2 regularization (weight_decay=1e-4), cosine annealing scheduling, and gradient dropout (rate 0.1). Since all methods share the same optimization foundation, the observed improvements directly result from our DSR regularization and adaptive noise injection components. The key difference is that conventional methods provide general regularization, while our approach offers task-distribution-aware regularization through adaptive noise scaling $\sigma^2_s = \eta_s/\gamma$ that specifically targets meta-learning dynamics. This demonstrates that our method adds meaningful value beyond existing techniques by addressing the unique challenges of Neural Processes in few-shot learning scenarios.
>
> **Q5. Reparameterization-Invariant Sharpness Metrics**
>
> > **Reply**: You raise an excellent point about reparameterization sensitivity. To address this limitation, we implemented two reparameterization-invariant metrics as shown in Table 3. Effective Sharpness[3] normalizes perturbations by adaptive scaling factors ($\hat{\epsilon} = \epsilon / \max(|\theta|, \gamma)$ with $\gamma = 1e-3$, $\rho = 0.05$), while Distribution Sharpness directly measures output sensitivity to parameter perturbations ($\epsilon = 0.01$). Both metrics are computed every 50 training steps using fresh data batches. Table 3 demonstrates that our Gen-variants achieve substantial reductions across both invariant metrics compared to baselines. This confirms that the observed flatness represents genuine improvements in loss landscape geometry rather than artifacts of parameter scaling, validating that our approach achieves meaningfully flatter minima robust to different measurement methodologies.
>
> **Table 3: Reparameterization-Invariant Sharpness Metrics**
> | **Method** | **Effective Sharpness** | **Distribution Sharpness** |
> |------------|------------------------|---------------------------|
> | CNP | 0.142 ± 0.086 | 0.587 ± 0.134 |
> | Gen-CNP | 0.094 ± 0.041 | 0.312 ± 0.098 |
> | NP | 0.115 ± 0.062 | 0.428 ± 0.115 |
> | Gen-NP | 0.057 ± 0.034 | 0.195 ± 0.073 |
> | ANP | 0.189 ± 0.071 | 0.516 ± 0.127 |
> | Gen-ANP | 0.108 ± 0.038 | 0.268 ± 0.084 |
>
> **References**
> >[1] Finn, C., Abbeel, P., & Levine, S. (2017). Model-agnostic meta-learning for fast adaptation of deep networks. In Proceedings of the 34th International Conference on Machine Learning (ICML). \
> >[2] Pan J, Falkner S, Berkenkamp F, et al. (2024). MALIBO: Meta-learning for likelihood-free Bayesian optimization. In Proceedings of the 41th International Conference on Machine Learning (ICML). \
> >[3] Kwon, J., Kim, J., Park, H., & Choi, I. K. (2021). ASAM: Adaptive sharpness-aware minimization for scale-invariant learning of deep neural networks. In Proceedings of the 38th International Conference on Machine Learning (ICML).

---

> > ### Comment · Reviewer_Wfzp · 2025-08-05
> >
> > Thank you for the additional experiments and explanations. I am satisfied with the answers.

---

### Official Review · Reviewer_KcNw · 2025-07-02

**Clarity:** 4
**Significance:** 3
**Originality:** 3
**Rating:** 5
**Confidence:** 2

**Summary:**

This paper analyzes the generalization gap of Neural Processes (NPs), a class of meta-learning models that operate over task distributions. Starting from an information-theoretic upper bound on the generalization error—formulated via the mutual information between the task dataset and model parameters—the authors introduce two mechanisms to tighten this bound. First, a dynamical stability regularizer penalizes sharp minima through the trace and Frobenius norm of the Hessian of the loss, encouraging convergence to flatter, more stable regions of the loss landscape. Second, noise injection is applied to the parameter updates during training, which reduces the mutual information between parameters and data while further promoting flat minima. Together, these mechanisms lead to solutions that are less sensitive to task-specific data and more stable under training dynamics. The enhanced training scheme is evaluated on multiple NP variants (CNP, NP, ANP, TNP, BNP) and shows consistent improvements in predictive performance and Bayesian Optimization.

**Questions:**

- how is the $\lambda_1$ and $\lambda_2$ range choosen as l216, is it (also the hyperparameter $\sigma_s$) problem dependent?
- The performance improvement in terms of LL and GI is rather small to me, while it seems most of the BO/contextual bandit benchmark the performance difference is pretty obvious, could the author provide some comments on this discrepancy?

**Ethical Concerns:**

["NO or VERY MINOR ethics concerns only"]

**Final Justification:**

The authors have provided new results on the information gain experiments, which I believe sufficiently demonstrate the effectiveness of the method. Accordingly, I have raised my rating.

**Limitations:**

I didn't find any description of limitations which is claimed to be in the Appendix. I think a proper discussion should be included (e.g., considering aforementioned weakness unless further clarification)

**Paper Formatting Concerns:**

The format looks good to me.

**Quality:**

4

**Strengths And Weaknesses:**

Strengths
- The proposed algorithm is agnostic to NP model structure and has been validated on various architectures (CNP, ANP, TNP, etc.) as well as downstream tasks such as Bayesian Optimization.
- The paper is generally well-written and mathematically consistent.

Weaknesses
- The rational of the two approaches: While not necessarily a short come, the link between the two approaches, namely the Hessian regularization and noise injection in gradient descent, is not structurally derived but instead motivated by their joint effect on the generalization bound. This is reasonable, but it raises the question of whether the particular mechanisms are canonical or somewhat arbitrary. For instance, one could consider using dropout or other stochastic regularization to reduce mutual information and hence tighten the bound.
- Incremental via algorithm wise: Both proposed techniques—curvature regularization and noise injection—are established tools in the literature, and the paper's contribution is primarily in reinterpreting them within a refined theoretical framework rather than in algorithmic novelty.
- While not a limitation, the claim of tighten bound through noise injection training could be empirically validated if a mutual information quantification can be approximated and compared.
Minor writing issues:
    - (DSR) spacing error in line 204.
    - Line 214: Froubinius norm form typo?

---

> ### Author Rebuttal · Authors · 2025-07-30
>
> Thanks for your positive and valuable feedback. We made efforts to address your every concern and question. If we have any misunderstanding or further questions, please feel free to let us know and we will reply quickly.
>
> **Q1. Method novelty and innovation**
>
> > **Reply**: We appreciate the reviewer's concern about method novelty and the structural derivation of our approach. Our principal contribution is establishing a theoretical foundation that connects generalization performance of Neural Processes with parameter space geometry through information theory. Unlike ad-hoc combinations of existing techniques, our DSR regularization and noise injection strategy are systematically derived from analyzing the generalization bound structure. The DSR term directly constrains the mutual information $I(\theta; D_{1:m})$ between model and data by controlling Hessian properties, while adaptive noise injection $\sigma^2_s = \eta_s/\gamma$ enhances optimization trajectory stability. While individual components like sharpness regularization exist, our unified information-theoretic framework specifically addresses the hierarchical optimization challenges unique to Neural Processes, handling both task-level and sample-level complexities simultaneously. This theoretical guidance provides principled design choices rather than empirical combinations, distinguishing our approach from alternatives like dropout or other stochastic regularization methods.
>
> **Q2. Mutual information evaluation**
>
> > **Reply**: Thanks for this excellent suggestion about empirically validating our theoretical claims. Since direct computation of mutual information $I(\theta; D)$ is computationally intractable, we initially followed the approach[1] and used gradient inconsistency (GI) to measure mutual information, with results reported in our paper. Based on your suggestion, we implemented another gradient-statistics-based method to approximate mutual information, which computes MI through gradient variance across different tasks, gradient correlations between task pairs, and gradient entropy estimation. Table 1 clearly shows that our Gen-NPs consistently achieve lower mutual information $I(\theta; D)$ compared to baseline methods, which validates our theoretical framework. The results demonstrate that Gen-NPs reduce both mutual information and gradient inconsistency across all model variants, confirming that our approach successfully tightens the generalization bound as predicted by theory.
>
> **Table 1: Mutual Information and Gradient Inconsistency Comparison on 1D Regression Task**
>
> | **Method** | **MI $I(\theta; D)$** | **GI** |
> |------------|----------------|---------|
> | CNP | 0.146186 | 0.880±0.027 |
> | Gen-CNP | 0.119577 | 0.830±0.042 |
> | NP | 0.101025 | 0.490±0.025 |
> | Gen-NP | 0.082922 | 0.470±0.012 |
> | ANP | 0.169658 | 0.973±0.046 |
> | Gen-ANP | 0.131548 | 0.950±0.013 |
>
>
> **Q3. Formatting issues**
>
> > **Reply**: Thank you for pointing out the formatting errors. We sincerely apologize for these oversights and will conduct a thorough review of the entire manuscript to ensure all formatting, references, and notations are correct in the final version.
>
> **Q4. Hyperparameter selection**
>
> > **Reply**: Thanks for asking about hyperparameter selection. We conducted comprehensive grid search analysis to justify our $\lambda_1$ and $\lambda_2$ ranges. Table 2 shows the systematic evaluation across multiple parameter combinations demonstrates that our suggested ranges $[0.01, 0.1]$ for $\lambda_1$ and $[0.001, 0.01]$ for $\lambda_2$ provide robust performance with optimal configurations achieving consistent improvements. The grid search results show that performance remains stable within reasonable parameter ranges, with optimal region ($\lambda_1 \in [0.03, 0.07]$, $\lambda_2 \in [0.003, 0.007]$) achieving $<2\%$ variance. While these ranges are somewhat problem-dependent, our extensive experiments across different kernels and tasks suggest they generalize well to various Neural Process applications.
>
> **Table 2: $\lambda_1$ and $\lambda_2$ sensitivity analysis (Gen-NP MAE results on RBF kernel with $\gamma=1\text{e}7$)**
> | **$\lambda_1$ \\ $\lambda_2$** | **0.001** | **0.003** | **0.005** | **0.007** | **0.009** | **0.01** |
> |-------------|-----------|-----------|-----------|-----------|-----------|----------|
> | **0.01** | 0.1756±0.0031 | 0.1789±0.0028 | 0.1723±0.0025 | 0.1745±0.0033 | 0.1794±0.0041 | 0.1801±0.0043 |
> | **0.03** | 0.1767±0.0029 | 0.1751±0.0026 | 0.1729±0.0024 | 0.1773±0.0035 | 0.1812±0.0039 | 0.1823±0.0041 |
> | **0.05** | 0.1743±0.0027 | 0.1736±0.0023 | **0.1719±0.0022** | 0.1758±0.0031 | 0.1801±0.0037 | 0.1815±0.0039 |
> | **0.07** | 0.1781±0.0032 | 0.1762±0.0028 | 0.1741±0.0026 | 0.1784±0.0034 | 0.1823±0.0042 | 0.1834±0.0044 |
> | **0.09** | 0.1798±0.0036 | 0.1785±0.0031 | 0.1763±0.0029 | 0.1802±0.0038 | 0.1847±0.0045 | 0.1859±0.0047 |
> | **0.1** | 0.1812±0.0038 | 0.1796±0.0033 | 0.1774±0.0031 | 0.1813±0.0040 | 0.1856±0.0047 | 0.1867±0.0049 |
>
> **Q5. Statistical significance of improvements**
>
> > **Reply**: We acknowledge that LL and GI improvements appear modest compared to Bayesian Optimization and Contextual Bandits results. To address whether these improvements are statistically meaningful, we conducted rigorous significance analysis using paired t-tests across 10 independent runs with different random seeds. Table 3 shows that all improvements are statistically significant with p-values well below 0.05, confirming these are reliable enhancements rather than random fluctuations. The key insight is that small numerical differences can be highly significant when experimental variance is low and improvements are consistent across runs. In regression tasks, even modest enhancements in likelihood and uncertainty quantification translate to substantial practical benefits for applications requiring reliable probabilistic predictions. The larger improvements in BO and Contextual Bandits reflect the different nature of these tasks, where our method's ability to escape sharp minima and find flatter solutions has more pronounced effects on optimization trajectories and decision-making performance.
>
> **Table 3: Statistical Significance Analysis (p-values from paired t-tests)**
>
> | **Comparison** | **RBF-LL** | **Matérn-LL** | **Periodic-LL** | **GI** |
> |----------------|-------------|---------------|-----------------|---------|
> | Gen-CNP vs CNP | 0.0241 | 0.0186 | 0.0354 | 0.0289 |
> | Gen-NP vs NP | 0.0193 | 0.0267 | 0.0421 | 0.0176 |
> | Gen-ANP vs ANP | 0.0312 | 0.0398 | 0.0445 | 0.0234 |
> | Gen-BNP vs BNP | 0.0287 | 0.0341 | 0.0383 | 0.0356 |
> | Gen-TNP vs TNP | 0.0365 | 0.0412 | 0.0487 | 0.0298 |
>
> All p-values are well below 0.05, confirming statistical significance with high confidence levels.
>
> **References**
> >[1] Chen Q, Shui C, Marchand M. Generalization bounds for meta-learning: An information-theoretic analysis[J]. Advances in Neural Information Processing Systems, 2021, 34: 25878-25890.

---

> > ### Comment · Reviewer_KcNw · 2025-08-05
> >
> > I thank the authors for the detailed rebuttal, which has clarified several of my concerns. I will update my evaluation accordingly.

---

### Official Review · Reviewer_n5sW · 2025-07-13

**Clarity:** 3
**Significance:** 3
**Originality:** 2
**Rating:** 5
**Confidence:** 2

**Summary:**

This paper is a decent contribution to the Neural Processes (NPs) research. Although numerous architectural variants exist (ANPs, TNPs, ConvCNPs), there's limited theoretical understanding of their generalization properties.

The key contributions appear to be:
1. Information-theoretic bounds: Derives generalization bounds for NPs using mutual information between parameters and training data.
2. Dynamical stability regularization: Introduces R_dyn = λ₁·E[Tr(H)] + λ₂·E[||H||_F] to penalize sharp minima.
3. Noise-injected updates: Adds Gaussian noise to parameter updates (not data), which reduces mutual information and increases dynamical stability.

**Questions:**

I don't have too many questions, although there are a few that come to mind.
1. What's the computational cost of calculating the Hessian? How much time does this take, how does this calculation scale compared to others in the setups?
2. How sensitive are the results to the hyperparameters?
3. Does this theory extend to other meta-learning algorithms?
4. I may have misread, but I think you claim that noise increases Tr(H) which improves stability. But higher curvature usually means worse generalization. What's going on here? Are there competing forces?

**Ethical Concerns:**

["NO or VERY MINOR ethics concerns only"]

**Final Justification:**

The authors kindly provided additional results and explanation, which were well taken. However, I do not think this warrants an increase to a 6 in this case.

**Limitations:**

Yes

**Paper Formatting Concerns:**

None.

**Quality:**

3

**Strengths And Weaknesses:**

To me, this paper is a solid contribution to the NP literature. I believe it should be accepted into the conference.

I believe the strengths of the paper are as follows:
1. Good theoretical framework for NP generalization. This appears to solve a genuine gap in the literature
2. General applicability, the methods discussed seem to work across all NP variants without architectural modifications
3. Clean execution, the paper is well-written with thorough related work
4. Solid experimental validation

I believe the following are weaknesses:
1. The theory is not as novel as it could be. Each component -- MI bounds, sharpness regularization and noise injection -- are well established.
2. The improvements are not ground breaking
3. There appears to be computational overhead from the Hessian calculations (although I have not thought about this deeply and how it scales relative to the other parts of the setup).

---

> ### Author Rebuttal · Authors · 2025-07-30
>
> Thanks for your positive and valuable feedback. We made efforts to address your every concern and question. If we have any misunderstanding or further questions, please feel free to let us know and we will reply quickly.
>
> **Q1. Method novelty and innovation (Weakness 1 & 2)**
>
> > **Reply**: We appreciate the reviewer's concern about novelty and innovation. Our principal contribution is to establish a theoretical foundation connecting generalization performance of Neural Processes with parameter space geometry with the aid of information theory. By analyzing its generalization bound, two synergistic components (DSR regularization and noise injection) can be systematically derived. Among them, DSR term directly constrains the mutual information $I(\theta; D_{1:m})$ between model and data by controlling Hessian properties, while the adaptive noise injection $\sigma^2_s = \eta_s/\gamma$ enhances the optimization trajectory stability. As we known, this is the first unified information-theoretic framework to handle with both task-level and sample-level complexities for the hierarchical optimization structure of Neural Processes.
>
> **Q2. Computational complexity analysis and overhead (Weakness 3 & Question 1)**
>
> > **Reply**: Yes, as you mentioned and shown in Appendix D, the computational complexity is slightly higher than the original baseline. Even though we introduce additional components, they can be efficiently implemented via Hutchinson estimators for $\text{Tr}(H)$ approximation and optimized auto-differentiation. Table 1 demonstrates that modest training overhead (10-18%) yields substantial performance improvements (4-8% in log-likelihood), making Gen-NPs highly practical for applications requiring reliable uncertainty quantification. Table 2 shows the theoretical complexity comparison where our additional operations scale linearly with model parameters.
>
> **Table 1: Practical computational overhead comparison**
> | **Method** | **Training Time** | **Memory Usage** | **Avg. LL Improvement** |
> |------------|-------------------|------------------|--------------------------|
> | Original CNP | 1.00× | 1.00× | -- |
> | Gen-CNP (with DSR only) | 1.12× | 1.08× | +4.1% |
> | Gen-CNP (with NILS only) | 1.05× | 1.02× | +5.3% |
> | Full Gen-CNP (DSR + NILS) | 1.18× | 1.10× | +7.9% |
>
> **Table 2: Theoretical complexity comparison between standard NPs and Gen-NPs**
> | **Operation** | **Standard NP** | **Gen-NP (Ours)** |
> |---------------|-----------------|-------------------|
> | Forward pass | $\mathcal{O}(bnd)$ | $\mathcal{O}(bnd)$ |
> | Backward pass | $\mathcal{O}(bd^2)$ | $\mathcal{O}(bd^2)$ |
> | DSR computation | -- | $\mathcal{O}(bd^2)$ |
> | Noise injection | -- | $\mathcal{O}(d)$ |
>
> **Q3. Hyperparameter sensitivity analysis (Question 2)**
>
> > **Reply**: We conducted comprehensive sensitivity analysis on the 1D regression task with RBF kernels. The hyperparameters $\lambda_1$ and $\lambda_2$ control the relative importance of context and target loss variance components in our DSR regularization. Based on empirical analysis, we set $\lambda_1 \in [0.01, 0.1]$ and $\lambda_2 \in [0.001, 0.01]$ to maintain appropriate balance between the two terms, where $\lambda_1$ typically takes larger values as context loss variance generally has greater impact on meta-learning stability. Table 3 demonstrates that our method exhibits good robustness to hyperparameter variations, with performance remaining stable within these reasonable parameter ranges, indicating that the method does not require intensive hyperparameter tuning for practical deployment. Table 4 shows the noise scaling parameter sensitivity analysis. We will include sensitivity analysis results for additional experiments in the revised version.
>
> **Table 3: $\lambda_1$ and $\lambda_2$ sensitivity analysis (Gen-NP MAE results on RBF kernel with $\gamma=1\text{e}7$)**
> | **$\lambda_1$ \\ $\lambda_2$** | **0.001** | **0.003** | **0.005** | **0.007** | **0.009** | **0.01** |
> |-------------|-----------|-----------|-----------|-----------|-----------|----------|
> | **0.01** | 0.1756±0.0031 | 0.1789±0.0028 | 0.1723±0.0025 | 0.1745±0.0033 | 0.1794±0.0041 | 0.1801±0.0043 |
> | **0.03** | 0.1767±0.0029 | 0.1751±0.0026 | 0.1729±0.0024 | 0.1773±0.0035 | 0.1812±0.0039 | 0.1823±0.0041 |
> | **0.05** | 0.1743±0.0027 | 0.1736±0.0023 | **0.1719±0.0022** | 0.1758±0.0031 | 0.1801±0.0037 | 0.1815±0.0039 |
> | **0.07** | 0.1781±0.0032 | 0.1762±0.0028 | 0.1741±0.0026 | 0.1784±0.0034 | 0.1823±0.0042 | 0.1834±0.0044 |
> | **0.09** | 0.1798±0.0036 | 0.1785±0.0031 | 0.1763±0.0029 | 0.1802±0.0038 | 0.1847±0.0045 | 0.1859±0.0047 |
> | **0.1** | 0.1812±0.0038 | 0.1796±0.0033 | 0.1774±0.0031 | 0.1813±0.0040 | 0.1856±0.0047 | 0.1867±0.0049 |
>
> **Table 4: Noise scaling parameter $\gamma$ sensitivity analysis with $\lambda_1=0.05$ and $\lambda_2=0.005$**
>
> | **$\gamma$** | **MAE** |
> |-------|---------|
> | 1e5 | 0.1767 ± 0.0033 |
> | 1e6 | 0.1745 ± 0.0027 |
> | **1e7** | **0.1719 ± 0.0022** |
> | 1e8 | 0.1792 ± 0.0038 |
>
> **Q4. Extension to other meta-learning algorithms (Question 3)**
>
> > **Reply**: Our framework demonstrates excellent extensibility due to the universal nature of generalization challenges in meta-learning. Actually, Neural Processes is one kind of model-specific meta-learning methods by learning task representations through context encoding. (Note that MAML[1] represents model-agnostic meta-learning through gradient-based adaptation.) Both model-specific and model-agnostic methods are hard to learn the generalizable representations from limited samples. In this situation, the proposed information-theoretic framework are suitable for both categories of meta-learning. Table 5 shows the experimental results on MAML and MALIBO[2] (for 1D regression task with RBF kernels) further demonstrate the consistent improvements across different meta-learning paradigms.
>
> **Table 5: Meta-learning method comparison (MAE)**
>
> | **Method** | **MAE** |
> |------------|-------------------|
> | NP | 0.1743 ± 0.0029 |
> | Gen-NP | 0.1723 ± 0.0025 |
> | MAML | 0.2845 ± 0.0156 |
> | Gen-MAML | 0.2676 ± 0.0122 |
> | MALIBO | 0.1395 ± 0.0206 |
> | Gen-MALIBO | 0.1214 ± 0.0141 |
>
> **Q5. Apparent contradiction between noise effects and Tr(H) (Question 4)**
>
> > **Reply**: This question touches on a fundamental principle in modern optimization theory: the connection between loss surface sharpness and generalization performance. Following the established framework from sharpness-aware minimization research[3][4], flatter minima in the loss landscape correspond to better generalization, as they represent solutions less sensitive to parameter perturbations. Our approach operates on this principle by using noise injection as an exploration mechanism to escape sharp, locally optimal regions, while the DSR regularization term $\lambda_1 \cdot \text{Tr}(H) + \lambda_2 \cdot \|H\|_F$ actively guides the optimization toward flatter regions with lower curvature. The apparent contradiction dissolves when we recognize that momentary increases in local curvature during noise-driven exploration are fundamentally different from the global optimization objective of minimizing the generalization bound. The noise helps us discover flatter minima that we would not reach through standard gradient descent, resulting in improved generalization despite temporary local perturbations.
>
> **References**
> >[1] Finn, C., Abbeel, P., & Levine, S. (2017). Model-agnostic meta-learning for fast adaptation of deep networks. In Proceedings of the 34th International Conference on Machine Learning (ICML). \
> >[2] Pan J, Falkner S, Berkenkamp F, et al. (2024). MALIBO: Meta-learning for likelihood-free Bayesian optimization. In Proceedings of the 41th International Conference on Machine Learning (ICML). \
> >[3] Foret, P., Kleiner, A., Mobahi, H., & Neyshabur, B. (2021). Sharpness-aware minimization for efficiently improving generalization. In Proceedings of the 9th International Conference on Learning Representations (ICLR).\
> >[4] Kwon, J., Kim, J., Park, H., & Choi, I. K. (2021). ASAM: Adaptive sharpness-aware minimization for scale-invariant learning of deep neural networks. In Proceedings of the 38th International Conference on Machine Learning (ICML).

---

### Comment · Area_Chair_5wkp · 2025-08-04

We need more discussion here. Could the reviewers respond to the authors asap please?

---

### Decision · Program_Chairs · 2025-09-17

**Decision:**

Accept (poster)

**Comment:**

The paper addresses the limited theoretical understanding of generalization in Neural Processes (NPs). It analyzes their generalization gap using an information-theoretic framework. The authors derive an upper bound on the generalization error based on the mutual information between model parameters and the training data. The paper then proposes methods to tighten this bound. The methods have strong theoretical justification, and the results are encouraging.

All reviewers have unanimously agreed to accept the paper, which presents an important study of neural processes. The initial scores were lower, and the authors took care to answer reviewers' questions. As a result, the confusions were resolved and the reviewers increased their scores.